# Extended Shine-Dalgarno motifs govern translation initiation in *Staphylococcus aureus*

Maximilian P. Kohl[1], Roberto Bahena-Ceron[2], Béatrice Chane-Woon-Ming[1], Maria Kompatscher[3], Matthias D. Erlacher [3], Charles Barchet[2], Ottilie von Loeffelholz[2], Pascale Romby[1], Bruno P. Klaholz [2] & Stefano Marzi [1]✉

Regulation of translation initiation is central to bacterial adaptation, but species-specific mechanisms remain poorly understood. We present high-resolution mapping of translation start sites in *Staphylococcus aureus*, revealing distinct features of initiation alongside numerous unannotated small ORFs. Our analysis, combined with cryo-EM of a native mRNA-ribosome complex, shows that *S. aureus* relies on extended, start codon proximal Shine-Dalgarno (SD) interactions, creating specificity against phylogenetically distant bacteria. Several natural *S. aureus* initiation sites are not correctly decoded by *E. coli* ribosomes. We identify new and conserved non-canonical start codons, whose regulatory initiation sites contain these characteristic extended SD sequence motifs. Finally, we characterize a novel example of uORF-mediated translational control in *S. aureus*, demonstrating that translation of a small leader peptide modulates expression of a key biofilm regulator. The described mechanism involves codon rarity, ribosome pausing, and arginine availability, linking nutrient sensing to biofilm formation in this major human pathogen.

*Staphylococcus aureus* (*S. aureus*) is a major opportunistic pathogen responsible for a wide range of community and hospital acquired infections. To survive in the host and ensure successful dissemination, it has evolved complex regulatory networks that respond to environmental cues by modulating the expression of numerous virulence factors[1]. While transcriptional regulation and sRNA-mediated control have been well characterized[2,3], alternative mechanisms acting at the level of translation remain largely unexplored. In bacteria, one of the key targets for regulation is the rate-limiting translation initiation step[4], which can be rapidly modulated in response to environmental cues and stress. Canonical translation initiation begins with the binding of the small ribosomal subunit (30S) to a translation initiation site (TIS) on a given mRNA. Within the TIS, the correct open reading frame (ORF) is defined by the presence of a start codon, usually AUG, GUG, or

UUG, which is decoded by formyl-methionyl initiator tRNA[fMet] in the ribosomal P-site. Start codon recognition represents a key step, resulting in a conformational change of the 30S initiation complex, enabling joining of the large ribosomal subunit and transitioning into the elongation phase of protein synthesis[5,6]. While both the efficiency as well as the accuracy of this decoding step are modulated by the presence of initiation factors[5,7], the initial ribosome recruitment is predominantly governed by the biophysical properties of an mRNA[8,9]. The accessibility of the TIS, particularly the presence of unstructured regions, plays a central role[9–11]. To facilitate and enhance ribosome binding, many mRNAs utilize Shine-Dalgarno (SD) sequence motifs, complementary to the 3′ end of the 16S rRNA, designated anti-SD (aSD)[12]. The SD-aSD interaction forms a short RNA helix at the ribosomal platform[13–16], where several ribosomal proteins may further

[1]Université de Strasbourg, CNRS, Architecture et Réactivité de l'ARN, Strasbourg, France. [2]Université de Strasbourg, CNRS, INSERM, Centre for Integrative Biology, Department of Integrated Structural Biology, IGBMC, Strasbourg, Illkirch, France. [3]Institute of RNA Biology and Genomics, Biocenter, Medical University of Innsbruck, Innsbruck, Austria. ✉e-mail: s.marzi@ibmc-cnrs.unistra.fr

modulate mRNA binding[4]. For example, *Escherichia coli* ribosomal protein bS1 is known to mediate affinity towards single-stranded mRNA regions[17,18] and contributes to the unwinding of secondary structures near the TIS[10]. Extensively studied in *E. coli*, the SD-aSD interaction is not strictly required for correct start site selection[19–21] but is thought to play a supportive role[19,22], although it can influence start codon recognition once the TIS is determined[23,24]. While our current understanding of bacterial translation initiation is largely based on studies in *E. coli*, growing evidence points to a functional diversification across bacterial species. Differences in SD usage, transcriptome architecture, rRNA sequence, and ribosomal protein composition suggest species-specific tuning of translation initiation mechanisms[20,25,26]. Of particular note, differences in the nature of ribosomal binding sites (RBS) have been recognized[27–29], but the underlying molecular basis remains unknown.

Precise mapping of TISs is vital to decipher the mechanistic divergence across bacteria, and to uncover the full coding potential of their genomes. However, small ORFs (sORFs) can elude conventional annotations and detection by proteomics[30–32]. Ribosome profiling (Ribo-seq) is the state-of-the-art technique to monitor translation dynamics globally[33], but its resolution in bacteria has historically lagged behind that achieved in eukaryotes. In eukaryotic systems, the use of RNase 1 for the generation of ribosome protected fragments (RPFs), in combination with harringtonine, enabled high-resolution mapping of initiation sites[34]. This approach has led to key discoveries such as the widespread translation of upstream ORFs (uORFs) functioning as regulators, which respond to stress and signaling pathways[34,35]. Furthermore, it led to the identification of several start sites utilizing non-canonical initiation triplets[36,37] and provided valuable insights into alternative initiation mechanisms and isoform diversity[38]. In bacteria, the main limitations stem in part from technical constraints in RPF generation. In comparison, RNase 1 has been underused due to its inactivity in *E. coli* extracts[39] and is typically replaced by micrococcal nuclease (MNase)[22,40–46], whose strong sequence bias limits obtainable resolution[47,48]. However, a recent study has shown that RNase 1 can generate high-resolution RPFs in *Listeria* and *Salmonella*[49]. Therefore, we employed RNase 1 in *S. aureus*, significantly enhancing reading frame resolution. We further coupled its use with the initiation-specific inhibitor Retapamulin (Ret), which has been used for global mapping of initiation sites in *E. coli*[50].

Our study now analyzes precise start codon selection on a translatome-wide scale in *S. aureus*, revealing accurate decoding of multiple sites, which could not be rationalized by the current understanding of bacterial translation initiation. Functional analysis shows that *E. coli* ribosomes fail to recognize the correct start codon when presented with specific natural *S. aureus* mRNAs, underscoring mechanistic differences in start site selection. Structural analysis by cryo electron microscopy (cryo-EM) of the *S. aureus* ribosome in complex with one such native *S. aureus* mRNA reveals start-codon proximal extended SD-aSD interactions. Such rigid SD-aSD helix formation on the platform of the 30S ribosomal subunit limits the flexibility observed in *E. coli* ribosomes, directly influencing start codon recognition. This new mechanistic understanding led us to assess novel candidate initiation sites in *S. aureus* with greater confidence and uncovered a diverse set of previously unrecognized regulators. Among these is the use of broadly conserved non-canonical start codons, which reduce initiation efficiency. The corresponding mRNAs all feature extended SD sequences, which aid recognition of the weak start codons. We also present the first comprehensive mapping of sORFs, including putative regulatory uORFs in *S. aureus*, and uncover a novel example of conditional regulation mediated by an unannotated leader peptide. Translation of this peptide directly influences the expression of the downstream transcription factor Rbf, a key regulator of biofilm formation[51,52]. Its uORF-dependent regulation is driven by a combination of codon rarity and arginine availability, leading to ribosome

pausing, consequent occlusion of the *rbf* RBS, and inhibition of translation initiation under arginine-limiting conditions. Collectively, our findings offer a precise mechanistic understanding of the pronounced differences in translation initiation, driven by the evolution of alternative RBS architectures across bacteria, and offer a broad perspective on translational control in a clinically relevant pathogen.

## Results

### RNase 1 Ribo-Ret reveals global *S. aureus* translation initiation at high-resolution

To capture a comprehensive snapshot of the *S. aureus* translatome, cells were grown towards the mid- to late exponential growth phase in rich medium, where virulence factors are highly expressed. Brief treatment of the cultures with Retapamulin (Ret) resulted in a pronounced arrest of initiating ribosomes (Fig. 1A, B). This was carefully monitored with non-digested controls, whose polysome profiles were analyzed in parallel (Fig. 1B). The addition of Ret resulted in a stark shift in the sucrose gradient from the polysome- towards the 70S fraction, indicating stalling during the initiation step. Likewise, treatment of the cell extracts with either MNase or RNase 1 enriched 70S peaks and resulted in a loss of polysomes, demonstrating a successful digestion for the generation of RPFs (Fig. 1B). Both MNase and RNase 1 digested samples, Ret-treated and non-treated, were processed for RPF-library preparation and next-generation sequencing.

Sequencing results were first analyzed to assess Ribo-seq quality to see whether RNase 1 could improve the resolution of Ribo-Ret in *S. aureus* (Fig. 1C, D). For either nuclease, in the absence of Ret, the read lengths formed a bimodal distribution with peaks at approximately 26–29 nt and 35–38 nt, respectively. Strikingly, the addition of Ret resulted in a strong shift towards the longer read population (Fig. 1D). The observed distribution is consistent with the approximately 28 nt protection from nuclease digestion, conferred by elongating ribosomes. The stark increase in read length upon Ret treatment can be rationalized by the additional protection that SD-aSD base pairings confer during initiation[53].

We next performed an average gene analysis of 3′ end densities, mapped across annotated ORFs, and confirmed a drastic enrichment of densities corresponding to initiating ribosomes in the Ret-treated samples (Fig. 1D). The peak positions, relative to annotated start codons, mapped at the expected 16 nt distance from the first nucleotide of the start codon (+17). As hypothesized, these peaks were sharper for RNase 1 and mapped predominantly (60%) to this position (Fig. 1C). This was in contrast to MNase, where preferential cleavage bias resulted in a broader distribution. Hence, RNase 1 Ribo-Ret provides global and high-resolution translation initiation snapshots.

### Discovery of new sORFs by high-resolution Ribo-Ret

While standard Ribo-seq data can be used to evaluate potential sORFs, the application of Ribo-Ret and the observed enrichment in ribosome density at putative start sites, provides stronger experimental support. We therefore leveraged our complementary Ribo-seq and Ribo-Ret datasets for comprehensive sORF discovery (Fig. 1E, F, G as well as Figs. S1 and S2). The pipeline used for sORF candidate filtering is outlined in Fig. S3. All considered ribosome densities were consistent between the two complementary datasets and RNase 1 allowed precise start codon assignment.

The obtained improved resolution was critical when Ribo-Ret densities could match to multiple potential sORFs. This was particularly the case for overlapping start codon sequences (e.g., AUGUG). A typical example is shown in Fig. 1E, where Ribo-Ret density mapped to a newly identified sORF (*sORF20*). In the RNase 1 dataset, the 3′ ends of Ribo-Ret reads accurately mapped at position +17 to an AUG start codon, while in the MNase dataset the corresponding peak falsely indicated translation from an overlapping GUG start codon. In-frame AUG start codon selection on *sORF20* mRNA was further biochemically

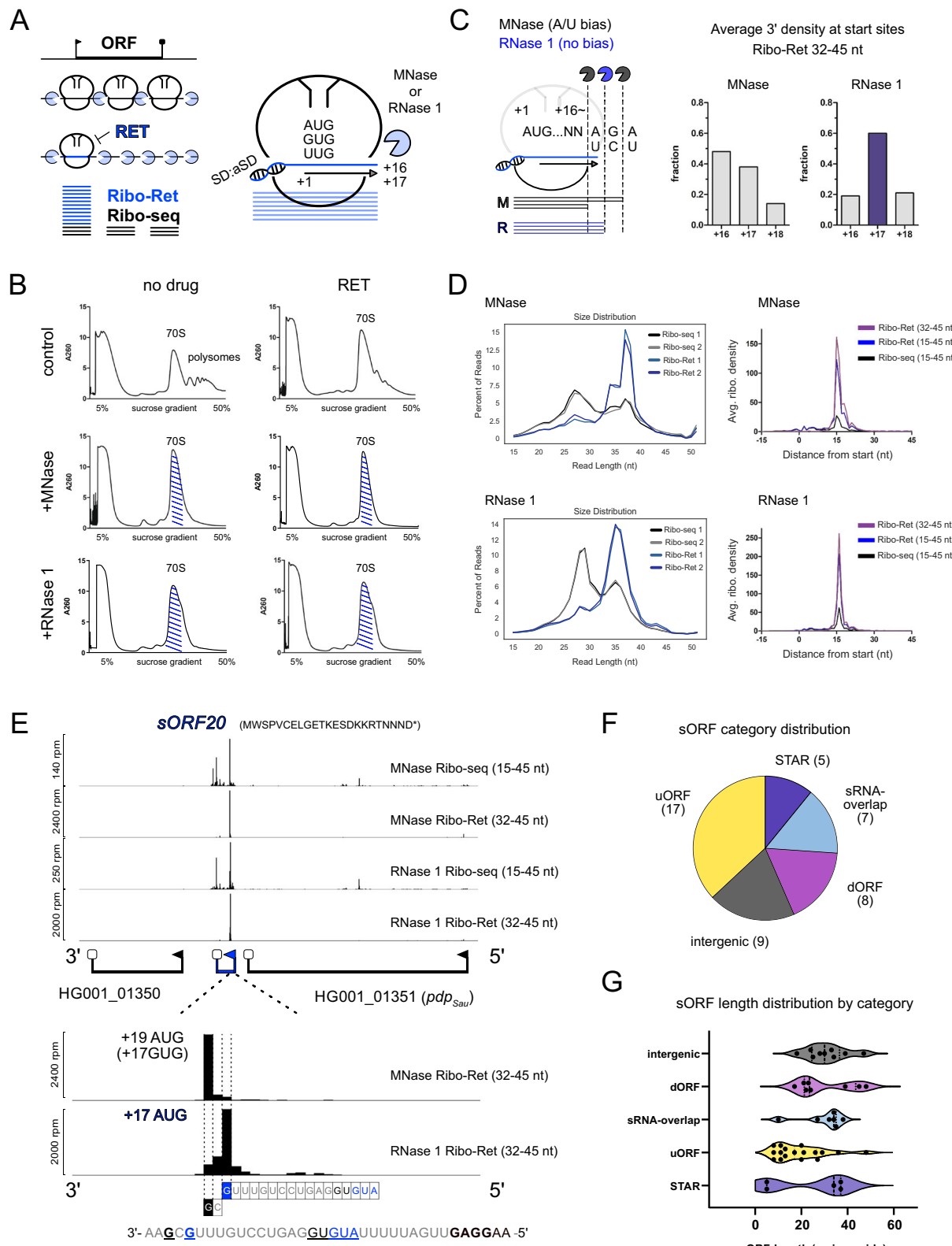

validated by toeprinting of an *S. aureus* 70S initiation complex stalled with Ret (Fig. S1).

Focusing our analysis on intergenic regions, we obtained a high confidence sORF map specific to our experimental growth conditions (Supplementary Data 1). In addition to previously identified functional small proteins (e.g., phenol soluble modulins[54]), our study led to the discovery of 46 unannotated sORFs. To our knowledge, only two of

these had previously been identified in a proteomics study[55], and one additional sORF had been reported to function in transcription antitermination within the isoleucine biosynthesis operon[56]. The other 43 sORFs represent bona fide novel candidates. Notably, several of these sORFs are predicted to encode small hydrophobic transmembrane helices, including *sORF26* and *sORF36*. Through toeprinting assays, analyzing *S. aureus* 30S initiation complex formation, their efficient

**Fig. 1 | RNase 1 Ribo-Ret enables high-resolution translation initiation profiling and sORF discovery in *S. aureus*. A** Schematic representation of Retapamulin (RET)-assisted ribosome profiling with MNase or RNase 1. **B** Sucrose density gradient profiles of Ribo-seq (no drug) and Ribo-Ret (RET) experiments with different nuclease treatment and mock treated controls. Monosome fractions isolated for analysis are highlighted in blue. **C** Schematic depiction of MNase cleavage bias and the obtained improved resolution by RNase 1 Ribo-Ret. Fractions indicate the relative amount of mapped read 3′ ends distributed between positions +16, +17, and +18 in the peak window of a global average gene analysis. **D** Read length

distribution of Ribo-seq and Ribo-Ret experiments obtained with either nuclease as well as plots of the average gene analysis used to assess start site resolution. Ret treated samples are highlighted in blue in both panels and in purple for the selective analysis of longer reads on the average gene plots. **E** Exemplary depiction of sORF discovery and start codon assignment in form of Ribo-seq and Ribo-Ret profiles of *sORF20*. Tracks show 3′ end Ribosome Protected Fragment (RPF) densities for MNase and RNase 1 with read length windows indicated (e.g., "RNase 1 Ribo-seq (15–45 nt)"). **F, G** Newly identified sORF categories and their comparative length distributions are shown. Also see Figs. S1–S3 and Supplementary Data 1.

ribosome recruitment was biochemically validated (Fig. S1C, D). Overall, 14 initiation sites were analyzed by toeprinting, confirming active start site selection (Supplementary Data 1).

Based on the genomic context, identified sORFs were classified into distinct categories (Fig. 1F). Several examples overlapped with previously annotated putative sRNA features (s*ORF6, 26, 30, 34, 35, 36, and 40*) and were therefore assigned an "sRNA overlap" category. We further identified nine sORFs located in intergenic regions, which were distinguished from sORFs embedded within 5′- or 3′ untranslated regions, designated as uORFs or downstream ORFs (dORFs), respectively. Among them, 17 sORFs could be classified as uORFs and 8 as dORFs. The *sORF20* (Fig. 1E), while possessing its own promoter, was categorized as dORF because it can be co-transcribed with its immediate upstream gene *Pdp_Sau* (HG001_01351), which encodes a recently identified anti-phage defense protein[57]. To complete our sORF categorization, we observed that several of the newly identified initiation sites shared sequence similarity and conserved RBS. These motifs stem from the conserved 5′ region of *S. aureus* repeat (STAR) motifs[58] (Fig. S2). We conclude that STAR motifs consistently recruit ribosomes when expressed. Among the 46 sORFs identified in our study, intergenic sORFs, dORFs, and sRNA overlapping sORFs tend to be longer than uORFs, which may act as regulatory leader peptides (Fig. 1G). In contrast, STAR-associated sORFs show considerable length variability. Finally, using a combination of BLASTN, tBLASTN, BLASTP, and targeted database searches, we identified high conservation for several sORFs, in particular for sORFs 15, 16, 18, 20, 24, 28, 29, 30, 34, 43, and 45, mostly restricted to Staphylococcaceae (Supplementary Data 1).

## Translation initiation in *S. aureus* follows distinct rules to define start codon selection

Our high-resolution RNase 1 Ribo-Ret analysis facilitated correction and re-annotation of approximately 40 coding sequences. These corrections ranged from minor adjustments, with accurate assignment of start codons among multiple in-frame possibilities, to more substantial changes as illustrated for *sarX* (Fig. S4). Importantly, while manually inspecting TISs, we observed unexpected patterns in start codon selection. Specifically, our RNase 1 data revealed cases of precise start codon definition despite potential ambiguity and competition. A typical example is the aureolysin (*aur*) mRNA, whose GUG start codon overlaps with an alternative AUG start codon in an AUGUG context, yet its correct initiation triplet is consistently selected (Fig. 2A, B). This was unexpected, as studies in *E. coli* have shown that a relatively shorter spacer length between the SD and the start codon is generally preferred[23], suggesting that the AUG in this context would be better positioned for initiation (Fig. 2A). Using our improved HG001 annotation, we analyzed spacing preferences and observed that on average, the aligned spacing between SD and start codon in *S. aureus* is increased by two to three nucleotides compared to *E. coli* (Fig. 2A). Although variations in spacer lengths among bacteria have previously been observed[27], they have not been linked to differences in start codon selection.

Because the identified initiation sites, such as *aur*, imply differences in the bacterial translation initiation mechanism, we hypothesized that several natural *S. aureus* mRNAs may not be accurately

decoded by *E. coli* ribosomes. To test this hypothesis, we performed cross-species experiments and studied start codon selection in the context of the *S. aureus aur* and *rlmB* initiation sites, which feature AUGUG overlaps and require recognition of an in-frame GUG start codon. Using purified *E. coli* or *S. aureus* 70S ribosomes supplied to a recombinant in vitro translation system (see "Methods"), we analyzed start codon usage by toeprinting of initiation complexes stalled in the presence of Ret. Consistent with our hypothesis, *E. coli* 70S preferentially selected the AUG positioned proximal to the SD helix, while only *S. aureus* ribosomes accurately decoded the in-frame GUG start codon (Fig. 2C), suggesting species-specific initiation preferences. To further validate these results in vivo, we constructed bi-cistronic dual reporter plasmids, in which either the AUG or GUG start codon of *aur* and *rlmB* was fused in-frame to a Firefly luciferase gene. The expression was normalized to the co-transcribed Renilla luciferase with a constant translation initiation context. Our data demonstrate that *S. aureus* strongly favors the annotated GUG start codon, whereas *E. coli* shows significantly higher expression from the closely spaced AUG in vivo (Fig. 2D).

## Extended SD-aSD base pairing confers evolutionarily distinct start codon selection

To better understand the mechanistic details of *S. aureus* translation initiation, we determined the cryo-EM structure of an *S. aureus* 70S ribosome initiation complex programmed with native *aur* mRNA. Consistent with our Ribo-Ret, toeprinting, and reporter gene data, the obtained cryo-EM structure reveals accurate decoding of the *aur* GUG start codon. Furthermore, the structure shows a characteristic SD-aSD helix and allows to closely examine the interactions, which are critical for accurate start codon positioning in the ribosomal P-site. Strikingly, beyond the canonical SD-aSD helix typically seen in bacteria[13–15], two additional Watson-Crick base pairs are found (Fig. 3A, B). These base pairs form between an AU dinucleotide following the canonical SD motif and nucleotides A1543 and U1544 on the 3′-region of the 16S rRNA. This extended duplex occupies a large region of the ribosomal platform, spanning further along the mRNA path, from uS11 towards uS2 (Fig. 3A, C). Positively charged residues of uS2 and bS18 contact the backbone of the 16S rRNA 3′ end and stabilize this shifted position of the SD-aSD helix (Fig. 3C). The additional base-paired AU dinucleotide corresponds to the increased spacing between SD and start codon in *S. aureus* that we found (Fig. 2A), and appears to strengthen the SD-aSD interactions, i.e., the sequence insertion is not a linker increase but rather it provides additional complementary base pairing opportunities.

Based on the extended Watson-Crick base pairing observed, we predicted that these interactions could be crucial for accurate start codon selection within the *aur* RBS. To assess whether this extended SD-aSD could help to resolve start site ambiguity, we performed toeprinting on variants of the *aur* mRNA (Fig. 3D, E). We first confirmed that *aur* start codon selection by *S. aureus* 70S is dependent on spacer length. Indeed, a single nucleotide insertion shifted decoding from the in-frame GUG to the competing AUG start codon, without affecting the *E. coli* ribosome's recognition of AUG (Fig. 3E). We next introduced mutations altering the start codon proximal AU dinucleotide involved in the extended SD-aSD pairing (Fig. 3A, B) to study its contribution to

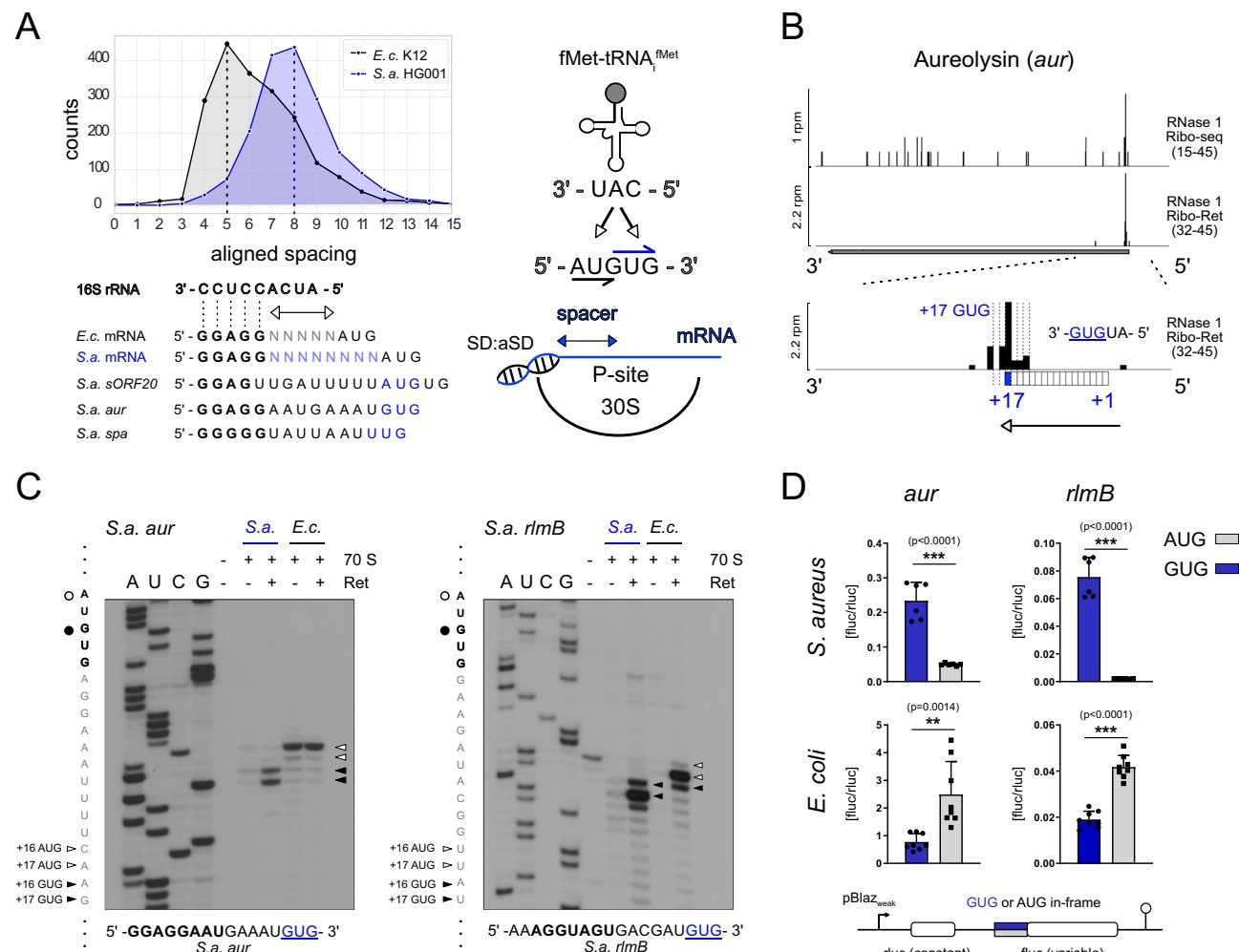

**Fig. 2 | High resolution Ribo-Ret reveals evolutionarily distinct start codon selection. A** Comparative analysis of aligned spacing distributions between *S. aureus* and *E. coli*. Exemplary *S. aureus* ribosome binding sites are displayed for reference and their in-frame start codons are highlighted in blue. Additionally, a schematic representation of SD-aSD directed localization of the start codon in the P-site is shown. **B** Ribo-seq and Ribo-Ret profiles of *S. aureus aur* mRNA showing RPF 3′ end density peaks, highlighting selection of an in-frame GUG start codon within an AUGUG overlap. **C** Toeprinting analysis of natural *S. aureus* mRNAs *aur* and *rlmB* decoded by *S. aureus* (*S.a.*) and *E. coli* (*E.c.*) 70S ribosomes. Sequencing lanes (A, U, C, G) and toeprint positions (arrows) are indicated accordingly. Ret denotes Retapamulin. **D** Results from in vivo dual luciferase assays measuring translation initiation efficiencies from in-frame AUG (gray) or GUG (blue) reporter fusions of *aur* and *rlmB* initiation sites in *S. aureus* and *E. coli*. Measurements corresponding to correct decoding of the natural in-frame GUG start codon are highlighted in blue. A schematic drawing of the dual luciferase reporter is depicted. Mean and standard deviation from six biological replicates for *S. aureus* and eight biological replicates for *E. coli* are shown. Two-tailed unpaired *t*-test, ∗∗∗*p* < 0.0001. Calculated *p*-values for significant differences in the comparison of reporter expression were <0.0001 for *rlmB* in *S. aureus* and *E. coli*, while for *aur* the *p*-values were <0.0001 in *S. aureus* and 0.0014 in *E. coli*.

start codon definition. A substitution with a CC pair abolished the GUG start codon recognition, and a CU dinucleotide failed to restore it. Partial recovery was observed with AC, while full restoration occurred with GU, indicating that a G:U wobble pair can substitute for A:U Watson-Crick interactions to maintain accurate start codon selection (Fig. 3E). In *E. coli*, all variants were decoded at the first initiation triplet (AUG) irrespective of spacer mutations, consistent with the absence of extended SD-aSD accommodation. The tested spacers placed AUG more optimally in the P-site of the *E. coli* ribosome, rendering start-site choice largely insensitive to local RBS changes that shift decoding in *S. aureus*.

The observed accessibility of nucleotides upstream of the canonical CCUCC sequence in the 3′ region of *S. aureus* 16S rRNA enables new possibilities for SD-aSD interactions that may contribute to accurate start site selection. In our structure with *aur* mRNA, A1546 of the aSD is flipped out, while the unpaired adenosine on the mRNA is maintained in a helical conformation, aided by stacking interactions

with both the previous and following residues as well as a single hydrogen bond that is formed with C1545 of the aSD (Fig. 3A, B). We hypothesize that alternative extended SD-aSD interactions could be accommodated in the given sequence space aiding translation initiation on natural mRNAs. A typical example is the *S. aureus rarA* mRNA, which contains a potentially ambiguous AUGUG start site but is accurately decoded despite a poorly defined canonical SD sequence (Fig. S5A). We hypothesized that, in such contexts, alternative energetically favorable SD-aSD duplexes could guide correct start codon placement in the P-site. This is supported by 30S toeprinting, which showed accurate GUG selection by *S. aureus* ribosomes, while *E. coli* 30S failed to efficiently decode either start codon (Fig. S5A). Because all studied examples so far had been in the context of AUGUG overlaps, we analyzed one further mRNA with two competing AUG start codons, designating alternative reading frames. Such an arrangement is found, for example, in the natural *proP* mRNA of *S. aureus*, where precise start codon selection is observed by Ribo-Ret and predicted to depend on

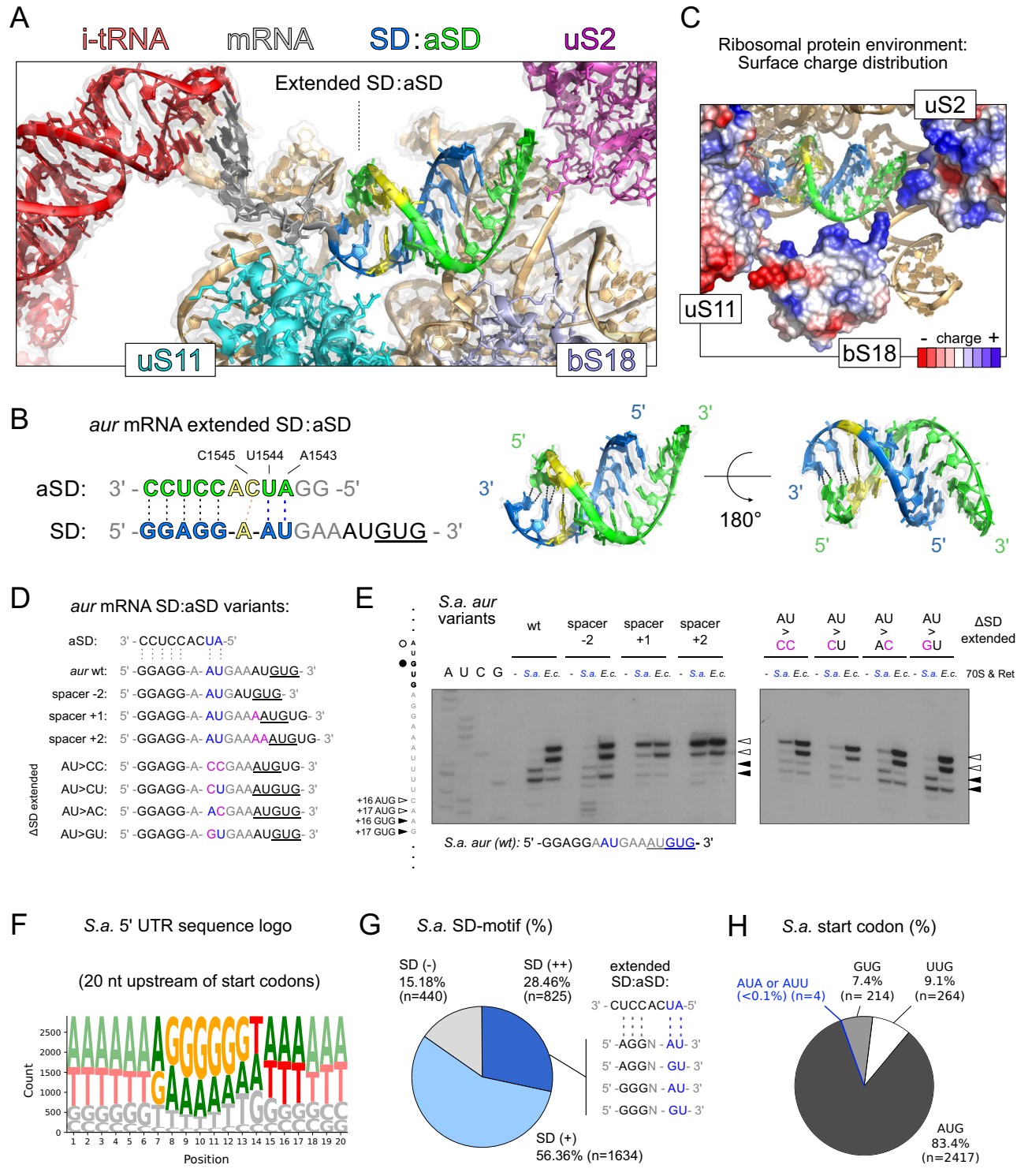

**A** i-tRNA mRNA SD:aSD uS2

Extended SD:aSD

uS11 bS18

**C** Ribosomal protein environment: Surface charge distribution

uS2 uS11 bS18

- charge +

**B** *aur* mRNA extended SD:aSD

C1545 U1544 A1543

aSD: 3' - CCUCC ACUA GG -5'

SD: 5' - GGAGG-A-AU GAAAUGUG - 3'

180°

**D** *aur* mRNA SD:aSD variants:

aSD: 3' - CCUCCACUA-5'

*aur* wt: 5' -GGAGG-A-AUGAAAUGUG- 3'

spacer -2: 5' -GGAGG-A-AUGAUGUG- 3'

spacer +1: 5' -GGAGG-A-AUGAAAAUGUG- 3'

spacer +2: 5' -GGAGG-A-AUGAAAAUGUG- 3'

ΔSD extended

AU>CC: 5' -GGAGG-A- CCGAAAUGUG- 3'

AU>CU: 5' -GGAGG-A- CUGAAAUGUG- 3'

AU>AC: 5' -GGAGG-A- ACGAAAUGUG- 3'

AU>GU: 5' -GGAGG-A- GUGAAAUGUG- 3'

**E** *S.a. aur* variants

wt | spacer -2 | spacer +1 | spacer +2 | AU>CC | AU>CU | AU>AC | AU>GU ΔSD extended

A U C G | *S.a. E.c.* ... 70S & Ret

+16 AUG ▷ C
+17 AUG ▷ A
+16 GUG ▶ A
+17 GUG ▶ G

*S.a. aur (wt):* 5' -GGAGGAAUGAAAUGUG- 3'

**F** *S.a.* 5' UTR sequence logo

(20 nt upstream of start codons)

Count / Position

**G** *S.a.* SD-motif (%)

SD (-) 15.18% (n=440)

SD (++) 28.46% (n=825)

extended SD:aSD:
3' - CUCCACUA-5'
5' -AGGN - AU- 3'
5' -AGGN - GU- 3'
5' -GGGN - AU- 3'
5' -GGGN - GU- 3'

SD (+) 56.36% (n=1634)

**H** *S.a.* start codon (%)

GUG 7.4% (n= 214)

UUG 9.1% (n=264)

AUA or AUU (<0.1%) (n=4)

AUG 83.4% (n=2417)

extended SD-aSD pairing (Fig. S5B). Consistent with our hypothesis, *S. aureus* 30S accurately decoded the second AUG start codon, whereas initiation complex formation was ambiguous between the two start sites for *E. coli*. Jointly, these findings indicate the important contribution of extended SD-aSD formation to start codon definition during translation initiation in *S. aureus*.

Interestingly, most *S. aureus* mRNAs feature AU-rich spacer sequences, which may similarly promote extended SD-aSD interactions, as revealed by global sequence motif analysis (Fig. 3F). While

we predict the presence of canonical SD-sequences in approximately 84% of *S. aureus* mRNAs based on computational SD motif search, our analysis indicates a conservative estimate of approximately 30% of initiation sites relying on extended SD-aSD base pairing (Fig. 3G). This prediction is based on the experimentally validated base pairing possibilities of the *aur* mRNA and does not include other alternative interactions, which we consider a likely possibility. Finalizing our computational analysis of *S. aureus* translation initiation contexts, we report an unusually high frequency of UUG start codons,

**Fig. 3 | Cryo-EM unveils an extended SD-aSD helix directing start codon selection. A, B** Cryo-EM map and model of the extended SD-aSD helix region within an *S. aureus* 70S initiation complex formed on the natural *aur* mRNA. The corresponding sequence is shown, highlighting nucleotides involved in Watson-Crick base pairing. Initiator tRNA is shown in red, the aSD sequence is displayed in green, the *aur* mRNA's SD motif is colored in blue and the mRNA's model towards the start codon is shown in gray. Start codon proximal extended interactions are displayed from an alternative angle on the right in (**B**). **C** Surface charge distribution of ribosomal proteins uS11, bS18, and uS2, forming contacts with the extended SD-aSD helix. Relative surface charge distribution of each individual protein is colored along a gradient from most positive (blue) to most negative (red) electrostatic potential. **D, E** Comparative *S. aureus* (*S.a.*) and *E coli* (*E.c.*) 70S toeprinting analysis

of *aur* mRNA sequence variants, analyzing the contribution of spacer length and extended SD-aSD base pairing to the decoding preferences of its natural AUGUG start codon overlap. Extended SD-aSD interactions of the wt sequence compared to sequence variants are shown in (**D**), while the toeprinting results are displayed in (**E**). On the side of the autoradiography, the mRNA sequencing lanes (A, U, C, G) are indicated, and the respective toeprint positions are highlighted by black (GUG) and white (AUG) arrows. **F** Global sequence motif analysis of *S. aureus* sequences upstream of annotated start codons. **G** Computationally predicted SD sequence usage among annotated *S. aureus* mRNAs and the percentage utilizing predicted extended SD-aSD base pairing interactions. **H** Global start codon use percentage among annotated *S. aureus* mRNAs.

approximately 9.1% of all annotated start sites, compared to around 2% in *E. coli* (Fig. 3H). Notably, this fraction may slightly overestimate the actual percentage of UUG start codon use, which is approximately 7% when considering only annotated ORFs with Ribo-Ret RPF expression peaks. In *S. aureus*, the remaining annotated start codons correspond to 83.4% to AUG and to 7.4% to GUG. Considering only ORFs expressed in our Ribo-Ret datasets, no strong difference was observed for extended or general SD motif usage between AUG, GUG, and UUG start codons. In this context, general SD motif usage increased to approximately 90% while extended SD motif usage slightly increased above 30%. Finally, we also identified four initiation sites, designating non-canonical start codons, which were further analyzed (Figs. 3H, 4 and S6).

### Newly identified non-canonical start codons regulate translation initiation in *S. aureus*

Several parameters modulate initiation efficiency, including start codon identity. We observed that Retapamulin enriched ribosome density at the non-canonical AUA start codon of *infC*, encoding IF3 (Fig. S6), which led us to hypothesize that additional examples could be found. During HG001 re-annotation, we identified two novel and conserved non-canonical start codons (Fig. 4A, B). Notably, for all three of these initiation sites, an extended SD sequence motif is predicted to direct the non-canonical start codon selection.

The first new-found site is the non-canonical AUU start codon of *cufC1*, encoding a 61 amino acid small protein, designated Conserved virulence factor C1. Although little is known about its function, it possesses a potential zinc finger domain and could be involved in transcriptional regulation (Fig. 4E). This gene is the first in an operon, which had been initially identified in a screen for the discovery of conserved virulence factors[59,60]. We could clearly assign an RNase 1 Ribo-Ret peak to position +17 in respect to the non-canonical AUU start codon (Fig. 4A). Intriguingly, a very strong and extended SD motif (Fig. 4C), as well as the non-canonical nature of the start codon are conserved among *Staphylococci*.

The second example we identified is the non-canonical AUA start codon of *ccpN*, encoding catabolite control protein N[61]. RNase 1 Ribo-Ret density mapped accurately at position +17 (Fig. 4B), clearly indicating its expression. Although CcpN's functional role in repressing gluconeogenesis has only been demonstrated in *B. subtilis*, both a 76% sequence similarity and a structural similarity according to an AlphaFold[62] prediction, suggest functional conservation (Fig. 4F). Consistently, the non-canonical start codon usage as well as the strong and extended SD-aSD base pairings with appropriate spacer length are conserved between *S. aureus* and *B. subtilis* (Fig. 4D).

Our discovery of non-canonical start codon usage suggests that both *cufC1*- and *ccpN* expression are strongly regulated at the level of translation initiation. To test this hypothesis, we constructed bicistronic dual luciferase reporter plasmids, fusing the respective translation initiation context with a firefly luciferase reporter gene and monitored translation efficiencies in vivo (Fig. 4G). We included the fourth non-canonical start site that we identified, an AUA start codon

of citrate synthase *citZ*, conserved in all *S. aureus* strains derived from the NCTC 8325 lineage, commonly used as laboratory strains (Fig. S6). Particularly because of CcpN's role in central metabolism, we analyzed translation initiation in different growth media conditions. As hypothesized, for all three genes, the synthesis of Firefly luciferase was strongly decreased by the use of a non-canonical start codon compared to cognate AUG mutants (Fig. 4G). For cells grown in poor media (RPMI), all non-canonical start sites conferred approximately 10% of the reporter expression compared to AUG. Surprisingly, we observed that this relative initiation efficiency was increased up to 30% in the case of *ccpN* AUA, when expressed in rich growth medium (BHI; Fig. 4G, H). In contrast, we observed only a modest relative increase for *citZ* AUA, and no media dependent effect for *cufC1* AUU. Therefore, the usage of non-canonical start codons reduced the yield of protein synthesis in vivo and particularly sensitized the expression of *ccpN* to metabolic changes.

### Ribo-Ret identified a variety of uORFs representing potential leader peptides

Our high-resolution Ribo-Ret analysis revealed initiation signatures for 17 uORFs with potential regulatory functions. These uORFs were selected for experimental validation based on four criteria: (i) proximity or overlap with the downstream TIS, (ii) virulence or physiological relevance of the downstream gene, (iii) strength/sharpness of Ribo-RET initiation peaks, and (iv) the presence of sequence features indicative of conditional control, such as rare codons, or intrinsic terminators. An overview of these small uORFs and their experimental validation is presented in Fig. 5A.

To assess their capacity for translation initiation, we performed toeprinting assays on eight selected candidates (Fig. 5A–D). Robust 30S initiation complex toeprint signals were observed for seven out of eight tested uORFs, while one candidate yielded only a faint signal. To further validate uORF translation in vivo, we generated translational fusions between the full uORF coding sequences and a Firefly luciferase reporter gene, constitutively expressed alongside a Renilla luciferase as an internal control. This assay confirmed in vivo translation of four uORF candidates. In contrast, the uORF with a faint toeprint signal (*sORF9*) exhibited only low translation efficiency in vivo (Fig. 5E).

We consider *sORF7* to be an intriguing leader peptide candidate due to its unusually Asn-rich sequence (Fig. 5B), but the function of the downstream gene product is unknown. Furthermore, *sORF15* and *sORF16* represent nearly identical uORFs (> 90% nucleotide sequence identity), located immediately upstream of the lysis modules within prophages Φ11 and Φ12, respectively (Fig. 5C). Notably, both are followed by predicted intrinsic transcription terminators, suggesting a potential cis-regulatory function at the mRNA level. Another interesting discovery is *sORF18*, a 21-amino acid uORF positioned directly upstream of the gene *gehA*, encoding lipase 1. We also note the presence of *sORF17* of similar length, located just downstream of *gehA* (Fig. 5D). While this arrangement may reflect independent small protein functions, a potential regulatory role of *sORF18* in *gehA* expression cannot be excluded.

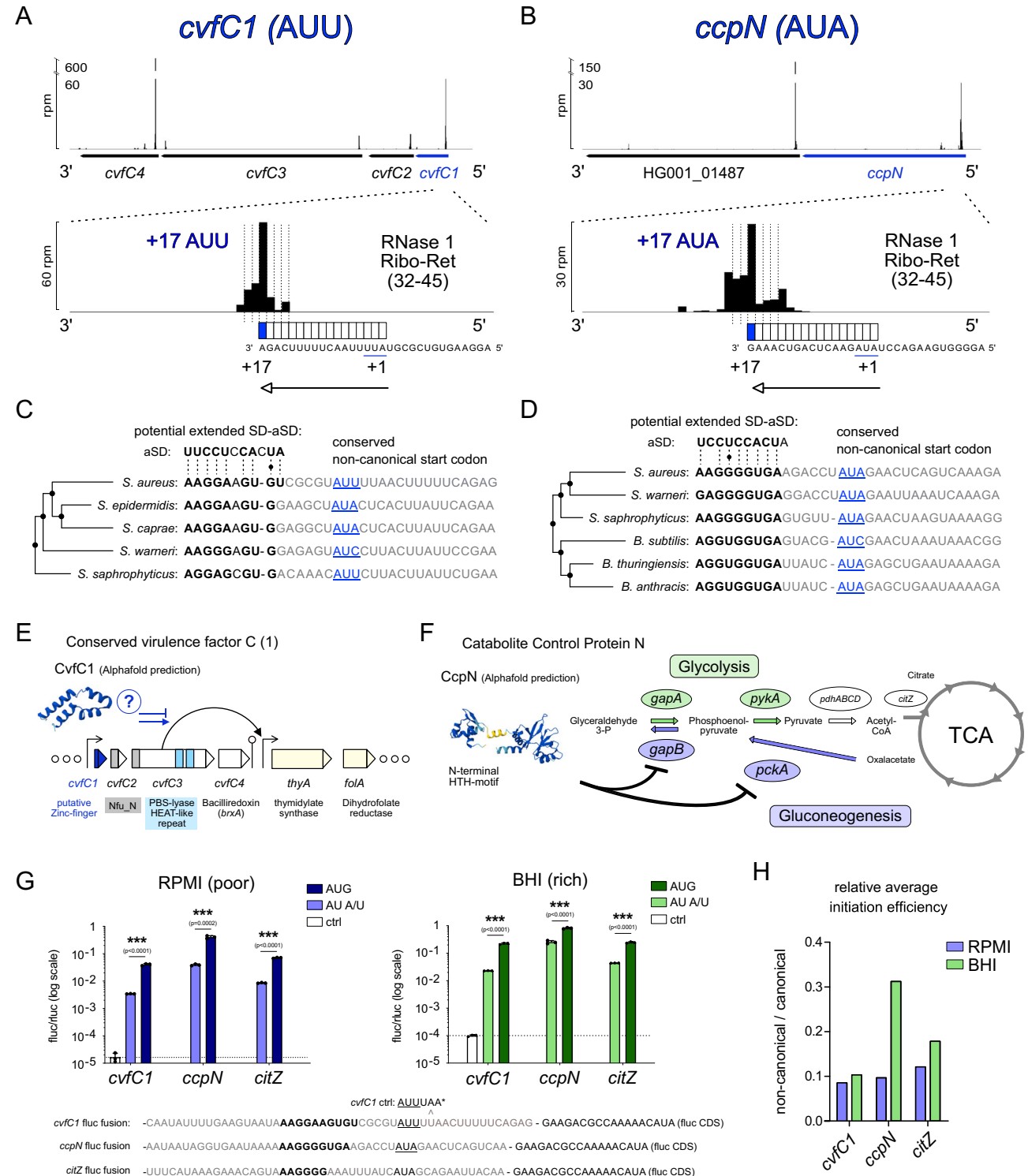

**Nature Communications** | (2026)17:2678

## A small uORF acts as a leader peptide and senses arginine restriction to control *rbf* translation

Short uORFs represent intriguing candidate leader peptides. We hypothesized that in some cases, their translation could act as a sensor for environmental cues or stresses, thereby regulating downstream gene expression. In addition to the Asn-rich *sORF7* (Fig. 5B), particularly *sORF5* drew our attention due to its location immediately upstream of the gene *rbf* (**r**egulator of **b**iofilm **f**ormation) (Fig. 6A). Notably, the luciferase reporter fusion of *sORF5* showed the highest in vivo expression levels among all tested candidates (Fig. 5E). Given its close proximity to and partial overlap with the *rbf* RBS, we investigated

its potential function as a regulatory leader peptide, hereafter referred to as *rbfL* (Fig. 6).

Although *rbfL* exhibited low expression during exponential planktonic growth, its discovery was enabled by the high sequencing coverage in our MNase data (Fig. 6A). Ribo-Ret density mapped to a GUG start codon with an appropriately spaced AGGGAAUGAU (potentially extended) SD motif and designated an eight amino acid small leader peptide (MVLKYRKR). Toeprinting confirmed efficient 30S initiation complex formation at the predicted *rbfL* GUG start codon, as well as at the downstream *rbf* start site (Fig. 6B). Because the CDS and stop codon of *rbfL* overlap the SD motif of *rbf*, we

**Fig. 4 | Ribo-Ret facilitates discovery of conserved non-canonical start codons.**
**A, B** Ribo-Ret profiles of *S. aureus* *cvfC1* (**A**) and *ccpN* (**B**) showing ribosomal density peaks and a zoomed focus on their respective initiation sites, highlighting the use of non-canonical start codons. Also see Fig. S6. **C, D** Sequence conservation of *cvfC1* (**C**) and *ccpN* (**D**) translation initiation contexts with extended SD-aSD base pairing and non-canonical start codon selection (in blue characters). **E** Schematic depiction of *cvfC1's* position in the *cvfC* operon. A putative Zinc finger fold of CvfC1 could indicate a contribution to the transcriptional regulation of *thyA* linked with CvfC3 expression. **F** Schematic depiction of CcpN's functional role as a transcriptional repressor of gluconeogenesis. CcpN's predicted structure via AlphaFold, containing an N-terminal Helix-turn-Helix (HTH) motif, is shown and black blunt arrows indicate repression of key gluconeogenic enzymes in the broader context of gluconeogenesis, glycolysis, and the tricarboxylic acid (TCA) cycle. **G** Results from in

vivo dual luciferase assays measuring translation initiation efficiencies from non-canonical (AUU or AUA) and canonical (AUG) start codon reporter fusions of *cvfC1*, *ccpN*, and *citZ* initiation sites in RPMI and BHI media. Measurements from variable Firefly luciferase reporter fusions are normalized by constant Renilla expression. For *cvfC1*, an additional plasmid variant with in-frame stop codon serves as negative control (ctrl). Results are displayed on a logarithmic scale as mean and standard deviation from three biological replicates. Two-tailed unpaired *t*-test, $*{*}{*}p < 0.0001$. Calculated p-values for significant differences in the comparison of AUG with AUA or AUU reporter expression were <0.0001 for *cvfC1 and citZ* in RPMI and BHI media, while for *ccpN* the *p*-values were <0.0001 in BHI and 0.0002 in RPMI. **H** Comparative plot of the average relative initiation efficiencies (fluc/rluc) of measurements displayed in (**G**), comparing non-canonical against canonical start codon usage in RPMI and BHI media.

---

hypothesized that *rbf* translation might be tightly coupled to efficient translation of its upstream leader peptide, *rbfL*. To test this hypothesis, we constructed dual luciferase reporter fusions, linking the *rbf* translation initiation region with Firefly luciferase and including its natural upstream sequence encompassing *rbfL*. As previously, Firefly luciferase was normalized using the bi-cistronic Renilla luciferase expression. An additional construct fused to the eight amino acid *rbfL* CDS was used to monitor leader peptide translation. These constructs allowed us to assess the influence of *rbfL* on downstream *rbf* translation across conditions. In rich BHI medium, mutating *rbfL's* GUG start- to a UAG stop codon only had a modest effect, resulting in a ~10% increase in *rbf* translation efficiency (Fig. 6C). However, under nutrient-poor conditions (RPMI medium), this effect of the leader peptide was markedly enhanced. *Rbf* reporter expression was strongly *rbfL*-dependent and ~fourfold higher in the absence of the leader peptide's start codon (Fig. 6D). Together, these findings suggest that nutrient availability modulates *rbf* translation efficiency in an *rbfL*-dependent manner.

Upon inspecting the *rbfL* coding sequence, we noticed that one of its two arginine codons was a particularly rare AGG codon (0.15% usage frequency). Both arginine codons (AGA and AGG) are decoded by the relatively scarce $tRNA^{Arg(UCU)}$[63]. We hypothesized that limited availability of the charged $tRNA^{Arg(UCU)}$ could constrain *rbfL* translation, thereby modulating downstream *rbf* expression. To test this, we repeated the dual luciferase reporter assays in RPMI medium supplemented with tenfold excess of L-Arg ($0.2 \rightarrow 2.2$/L). Strikingly, this supplementation increased *rbfL* translation efficiency three-fold (relative to the Renilla luciferase) and fully relieved its repressing effect on *rbf* translation (Fig. 6E). To further test the role of the rare arginine (AGG) codon, we substituted it with either a glycine (GGG) or a glutamate (AGA) codon, ensuring that neither mutation disrupted SD-aSD base pairing with the overlapping *rbf* RBS (Fig. 6A). Both mutations abolished *rbfL*-dependent repression and rendered *rbf* insensitive to L-Arg supplementation (Fig. S7D). Because these changes increased *rbf* translation irrespective of uORF translation or arginine levels, they uncoupled *rbf* from *rbfL*, consistent with *rbfL*-dependent control, but not alone proving the rare arginine codon's necessity. Substitution with another arginine codon (e.g., CGG) could, in theory, preserve the SD-like sequence but would introduce an even rarer codon decoded by the $tRNA^{Arg(CCG)}$, the least abundant tRNA in *S. aureus*[63]. We therefore increased the supply of the decoding tRNA by introducing a plasmid-borne *argU* gene (encoding $tRNA^{Arg(UCU)}$) under its native promoter in the wild-type reporter context. Consistently, elevating $tRNA^{Arg(UCU)}$ levels partially alleviated *rbfL*-dependent repression. The *rbf* expression in the absence of *rbfL's* start codon was ~1.7-fold higher than with it, versus a ~ fourfold difference without *argU* (Fig. 6F, G).

To further study whether a slow decoding and pausing at the rare arginine codons of *rbfL* contributed to *rbf* regulation, we performed additional toeprinting experiments following ribosome progression during in vitro translation of *rbfL* and its mutant sequence variants

(Fig. S7A, B). We first analyzed the co-translational toeprinting pattern of *S. aureus* 70S on the native *rbfL* sequence. Consistent with co-translational pauses, *rbfL* in vitro translation yielded several toeprint bands, which were lost upon treatment with Ret or the termination inhibitor Apidaecin-137. Among them, positions +13 of both the AGA and the AGG arginine codons displayed prominent bands, indicating their slow decoding in the ribosomal A-site. To demonstrate that these pauses were dependent on the AGG and AGA codons, alanine-scanning mutations were performed on *rbfL's* RKR sequence motif. Consistently, the observed pauses were lost when mutating the respective arginine codons to alanine, demonstrating their specific contribution to co-translational pausing (Fig. S7B). Notably, these pauses arise in vitro under optimal conditions and are expected to be drastically exacerbated under conditions that lower tRNA charging levels in vivo, such as nutrient limitation.

Collectively, these findings support a mechanism of RBS occlusion, in which slow decoding of the rare AGA/AGG codons in *rbfL* causes ribosome pausing that impedes translation initiation at the downstream *rbf* start codon (Fig. 6H). Our data strongly imply that arginine availability is the primary cue sensed by *rbfL* through the charging status of $tRNA^{Arg(UCU)}$, directly influencing the extent of pausing during *rbfL* translation. Based on this conclusion, we performed a final experiment, in which we quantified the effect of exogenous L-Arg supplementation on *S. aureus* biofilm formation in poor media (Fig. S7C). In line with this mechanism, we quantified biofilm formation in poor medium with or without exogenous L-Arg. After 48 h of static growth, crystal violet staining showed significantly increased biofilm in RPMI supplemented with L-Arg, while planktonic growth in the supernatant decreased, indicating a specific promotion of biofilm formation rather than a general growth effect (Fig. S7C).

## Discussion

Regulation of translation initiation is central to bacterial adaptation, but species-specific mechanisms remain poorly understood. In this study, we investigated translation initiation and its regulation in *S. aureus* by high-resolution initiation profiling with Ribo-Ret[50]. While RNase 1 has previously been used for standard Ribo-seq in *Listeria* and *Salmonella*[49], our work provides the first high-resolution, genome-wide dataset of TISs in bacteria. Although Ribo-Ret has gained popularity predominantly for sORF discovery[64–67], we exploited its high precision obtained with RNase 1 to explore translation initiation mechanisms at single-nucleotide resolution. By combining these profiling data with cryo-EM analysis, using the endogenous *S. aureus* aureolysin mRNA, a major virulence factor, we uncovered the structural basis for species-dependent start codon selection, revealing that *S. aureus* 70S ribosomes accommodate a shifted and extended SD-aSD duplex with additional base pairing as compared to *E. coli*. This structural stabilization rationalizes previously noticed differences in preferred spacing between the SD and start codon across bacteria[27,68] and provides a mechanistic explanation for divergent start codon selection in *S. aureus*. In bacteria, the SD-aSD helix provides an

## A  *S. aureus* HG001 small uORFs identified by Ribo-Ret

| sORF ID | uORF ID | distance (stop to start) | start codon | toeprint | conservation | sequence |
|---|---|---|---|---|---|---|
| sORF 3 | *gpmA* uORF | 110 nt | AUG | n.d. | in *S. aureus* | MYLMSSDN* |
| sORF 4 | *recX* uORF | 27 nt | UUG | n.d. | in *S. aureus* | MNNKRVLY* |
| **sORF 5** | *rbf* uORF | 4 nt | GUG | strong | in *S. aureus* | MVLKYRKR* |
| **sORF 7** | HG001_02379 uORF | 34 nt | AUG | strong | in *S. aureus* | MNLVNYNGNNN* |
| sORF 8 | HG001_01610 uORF | 111 nt | AUG | n.d. | in *S. aureus* | MLKVILNTLFI* |
| **sORF 9** | *pbp4* uORF | 39 nt | AUG | faint | in *S. aureus* | MISVSRNNHCF* |
| sORF 10 | HG001_04219 uORF | 39 nt | AUG | n.d. | in some *S.a.* strains | MDNINKNEKFRVL* |
| sORF 11 | HG001_02149 uORF | 32 nt | AUG | n.d. | in *S. aureus* | MECNVYIVCITDK* |
| sORF 13 | HG001_02012 uORF | - 4 nt (overlap) | AUG | n.d. | in several *S.a.* prophages (with sequence variation) | MKIKTASIEVEKVEVVV* |
| **sORF 15** | Φ11 lysis module uORF | 59 nt | AUG | strong | in several *S.a.* prophages (with sequence variation) | MLKGILGYSFWACFWFGKCK* |
| **sORF 16** | Φ12 lysis module uORF | 58 nt | AUG | strong | in several *S.a.* prophages (with sequence variation) | MLKGILGYSFWSCFWFSKCK* |
| **sORF 18** | *gehA* (*Sal1*) uORF | 24 nt | AUG | strong | broadly in *Staphylococci* | MSAWLSKLFEFIPRIIINLFI* |
| sORF 24 | *ilvD* uORF (Kaiser et al. 2018) | 158 nt | AUG | n.d. | broadly in *Staphylococci* | MLNQYTEHQPTTSNIIILLYSLGLER* |
| sORF 25 | HG001_04121 uORF | 29 nt | AUG | strong | in some *S.a.* strains | MKRETKQKISFCLSIGIFILVILLLIF* |
| sORF 28 | HG001_01162 uORF | 26 nt | AUG | n.d. | in some *S.a.* strains | MTRELRKKLTLYLNIATLILFIINLTRKK* |
| sORF 37 | HG001_01842 uORF | 1 nt | AUG | n.d. | in some staphylococcal prophages | MTEQMYLILFLLSLPLLLFIGRKTHFY CLDKKNGRR* |
| sORF 45 | HG001_00193 uORF | 24 nt | AUG | strong | small protein homologs in some Bacillus species e.g., *B. clausii* | MVESMLTFMLGPLRQITDFYMEHLLV SNSIVIAGYFATGIFKKKKVVN* |

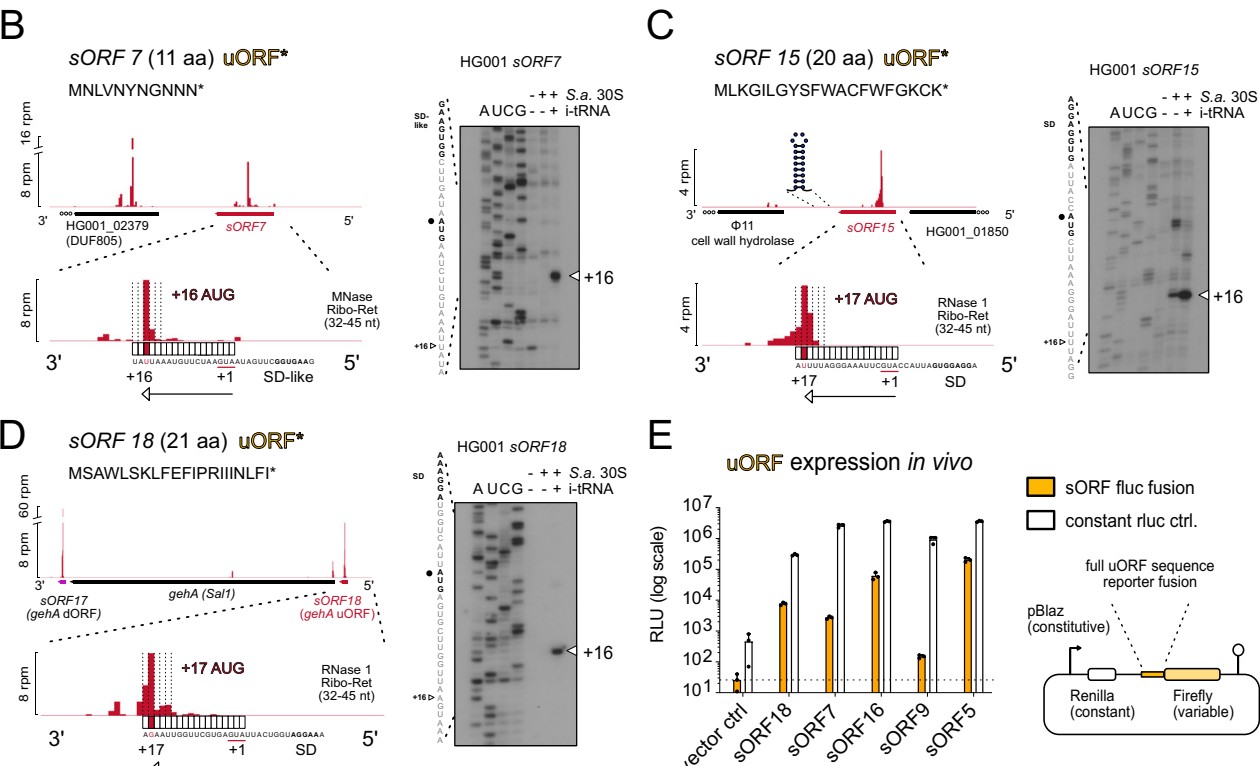

**Fig. 5 | Small uORFs represent candidate regulatory leader peptides. A** Selected sORF table displaying novel small uORF candidates. Their sORF- and uORF IDs, defined by respective downstream gene annotations, their start codons, coding sequences, distances to downstream genes, and conservation are shown. An additional column indicates whether toeprint analyses were performed and if the obtained signal was strong, faint, or not determined (n.d.). uORFs that were further experimentally analyzed are highlighted in bold. **B–D** Ribo-Ret profiles of *S. aureus* sORF7 (**B**), sORF15 (**C**), and sORF18 (**D**), showing ribosomal density peaks and a zoomed focus on their respective initiation sites. Toeprinting analyses validating strong 30S initiation complex formation are shown. The corresponding sequencing lanes (A, U, C, G) and the toeprint positions (+16) are indicated. **E** Selected validation of sORF expression in vivo using dual luciferase reporter fusions of candidate uORFs (*sORF18, sORF7, sORF16, sORF9, and sORF5*). Firefly luciferase signals from variable sORF reporter fusions are highlighted in orange, whereas measurements from constant Renilla luciferase are shown in white. Absolute measurements (relative light units, RLU) are displayed on a logarithmic scale as mean and standard deviation from three biological replicates. A sketch of the bi-cistronic reporter plasmid architecture is shown.

anchoring point for the mRNA during its accommodation on the ribosome. An optimal distance between the SD-aSD helix and the start codon ensures proper positioning of the start codon in the ribosomal P-site, facilitating its recognition by the initiator tRNA. In *S. aureus*, the

extended SD-aSD helix influences how start codons are presented in the ribosomal P-site (Fig. 3). The rigidity introduced by the additional start codon proximal interactions limits mRNA flexibility. Together with the shifted position of the extended SD-aSD helix on the

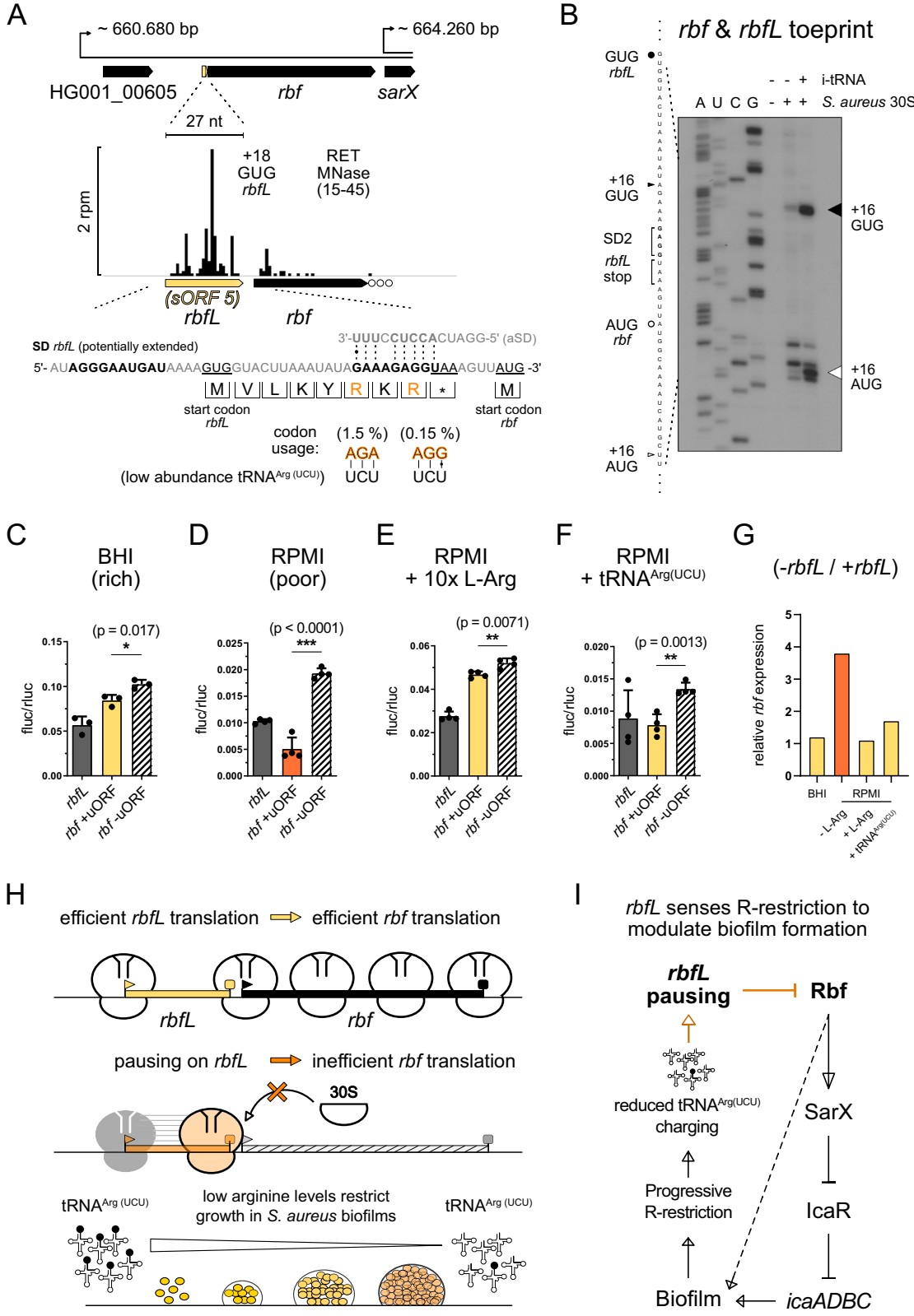

ribosomal platform, they contribute to the specific optimal spacer length that is required to correctly position the start codon in the P-site (Fig. 2). In *E. coli*, a shorter SD-aSD helix with a slightly different position on the platform, requires shorter spacing for optimal localization of the start codon. In the absence of alternative, competing start codons, suboptimal spacer length incurs only a modest energetic cost, lowering initiation efficiency[24]. However, when two potential start

codons are present in the same RBS, the optimally spaced start codon will be preferred over the suboptimally positioned alternative[23]. Strikingly, in these situations, the distinct spacer preferences observed among different bacterial species can lead to species-dependent decoding outcomes. This is demonstrated by our characterization of start codon selection in the context of *aur* and *rlmB* mRNAs, where *S. aureus* and *E. coli* initiate translation of different ORFs (Fig. 2C, D). The

**Fig. 6 | *rbfL* is a novel leader peptide, sensing arginine restriction and exerting translational control over a transcriptional regulator of biofilm formation.**
**A** Genomic organization of the *rbf* gene locus with its newly discovered leader peptide *rbfL* is depicted alongside a zoomed in MNase Ribo-Ret profile of corresponding ribosomal density peaks. The translation initiation sites of *rbfL* and *rbf* with their respective nucleotide sequence context are shown. SD motifs and start/stop codons are highlighted bold and underlined, respectively. *rbfL* peptide translation is displayed, emphasizing decoding of rare arginine codons (AGA and AGG). **B** Toeprint analysis demonstrating efficient 30S initiation complex formation at the start sites of *rbf* and *rbfL* in vitro. The corresponding sequencing lanes (A, U, C, G) and toeprint positions (+16) are indicated. **C–F** Results from in vivo dual luciferase assays measuring translation initiation efficiencies from *rbfL* and *rbf* reporter fusions in *S. aureus*. Culture conditions, such as growth media or L-Arg supplementation are indicated. Expression of *rbf* from its native sequence context in the presence of *rbfL* is highlighted in yellow, whereas *rbfL* start codon mutations (GUG > UAG) are indicated by striped white bars. The strong regulatory effect of *rbfL* in RPMI is emphasized by the change from yellow to orange color for the *rbf* reporter fusion in (**D**). Reporter data obtained in the presence of plasmid encoded

tRNA$^{Arg(UCU)}$ expression is shown in (**F**). Measurements from Firefly luciferase reporter fusions were normalized by bi-cistronic Renilla luciferase expression. Mean and standard deviation are shown from three biological replicates in (**C**) and from four biological replicates in (**D–F**). Two-tailed unpaired *t*-test, ∗∗*p* < 0.0001. Calculated *p*-values for significant differences in the comparison of *rbf* reporter expression in absence or presence of uORF translation were 0.017 in BHI (**C**), <0.0001 in RPMI (**D**), 0.0071 in RPMI with L-Arg supplementation (**E**), and 0.0013 in RPMI with tRNA$^{Arg(UCU)}$ expression (**F**). **G** Comparative plot of the relative *rbf* reporter expression levels displayed in (**C–F**), normalizing the expression of the *rbf* reporter fusion with *rbfL* start codon mutation (*rbf* -uORF = −*rbfL*) by *rbf* expression in the presence of *rbfL* (*rbf* + uORF = +*rbfL*) across different experimental conditions. **H** Graphical model of *rbf* translation regulation conferred by the *rbfL* leader peptide. **I** Schematic model linking a possible sensory role of *rbfL* translation to metabolic feedback control over *S. aureus* biofilm formation. Rbf indirectly represses the synthesis of *icaR* through SarX, which results in derepression of *icaADBC* to increase the production of poly-N-acetylglucosamine, a major component of biofilm[51,52].

---

potential for extended SD-aSD base pairing implies that a comparatively larger portion of the 16S rRNA is accessible for interaction with the mRNA, thereby allowing complementary sequences beyond canonical motifs to function as effective SD elements. In species capable of forming extended SD-aSD helices, the definition of SD motifs becomes less restrictive, encompassing a broader range of sequences. For example, in the *rarA* and *proP* mRNAs, such atypical alignments may underlie the accurate and efficient start codon recognition observed in *S. aureus*, but not in *E. coli* 30S initiation complexes (Fig. S5). While we do not define the full spectrum of these interactions in *S. aureus*, our data clearly show that extended SD-aSD pairing is a key determinant of start codon selection and contributes to evolutionary distinct patterns of ORF recognition. These findings underscore the potential to engineer mRNAs, in which overlapping ORFs in distinct reading frames are selectively translated by different bacterial species. This concept has broad implications for biomedical and biotechnological applications, including precise control of protein expression, species-specific translational switches in synthetic biology, and the targeted design of pathogen-specific antibacterial strategies.

Given the conserved 3′-end sequence of the 16S rRNA between *S. aureus* and *E. coli*, other features must account for the mechanistic divergence that enables extended SD-aSD formation and distinct start codon selection (Figs. 2 and 3). One notable difference is the presence of ribosomal protein bS21 near the SD-aSD helix in *E. coli*, which would sterically hinder the extended interaction observed in *S. aureus*. In Firmicutes, bS21 is relatively shorter and lacks the C-terminal residues needed to anchor it to the 16S rRNA[69], consistent with the absence of a corresponding EM-density in our structure (Fig. 3). Notably, bS21 has previously been linked to evolutionary translation initiation adaptations[26]. Another important difference is the absence of ribosome-anchored protein bS1 in *S. aureus*[70]. While bS1 is essential in *E. coli* and facilitates initiation by promoting mRNA loading and resolving structured ribosome-binding sites[10,71], its function in *S. aureus* is poorly understood and appears to diverge substantially. Further work will be required to clarify the respective contributions of bS21 and bS1 to translation initiation in *S. aureus*. Which evolutionary pressure might have driven the selection of extended SD-aSD base pairing? We hypothesize that this adaptation could be linked to genome composition, specifically, the high AT content in species like *S. aureus*, and the need to adapt their translation initiation strategy accordingly. In *E. coli*, which has a more balanced GC content, ribosomes predominantly rely on an unstructured region near the start codon for recruitment, and the SD-aSD pairing is not critically required[19]. In contrast, AT-rich genomes, like those of many Firmicutes, may rely more strongly on stable SD-aSD interactions. This could be due to the generally lower mRNA secondary structure in these

organisms, making it more difficult to define initiation sites by their unstructured nature[72]. Consistently, when we predicted average local RNA secondary structure folding propensities across annotated ORFs in *S. aureus* and *E. coli*, only the latter displayed significantly more positive minimum free energy around the start codons.

Our findings helped define which RBS sequences strongly promote translation initiation, hence identifying sORF candidates with greater confidence and enabling the discovery of novel non-canonical start codons. In *S. aureus*, translation attenuation via non-canonical start codons has not previously been described. This mechanism typically keeps gene expression low through inefficient decoding in the ribosomal P-site, a process modulated by IF3[73,74], which itself uses an AUA start codon in *S. aureus*, likely enabling auto-regulation as seen in *E. coli*[75]. Using Ribo-Ret, we identified two additional conserved non-canonical start codons limiting initiation efficiency of *S. aureus* *cvfC1* and *ccpN* mRNAs. The strong conservation of *ccpN*'s non-canonical start codon may be linked to CcpN's function as a transcriptional repressor of gluconeogenesis[61,76]. Strikingly, we observed that *ccpN's* initiation efficiency varies in a growth media-dependent manner. Under nutrient-rich conditions, translation initiation increases, hinting at a potential metabolic feedback control mechanism. Efficient *ccpN* translation could repress gluconeogenesis when nutrients are abundant, and inefficient translation would relieve this repression under starvation. Although a clear mechanistic understanding is lacking for this observation, we hypothesize that it could be linked to the connection between nutrient availability and regulation of translation initiation through one-carbon metabolism[77].

We have identified 43 novel sORFs and validated several initiation sites using toeprinting and reporter fusions (Supplementary Data 1). Especially, short uORFs are challenging to detect with standard methods like mass spectrometry or regular Ribo-seq. In these cases, Ribo-Ret's strong enrichment at start sites determined at high-resolution was critical to enable their identification. While our focus was primarily on regulatory uORFs, several of the newly-identified small proteins (e.g., *sORF25, 26, 28, 30, 36,* and *45*) are predicted to adopt hydrophobic transmembrane helices, suggesting possible functional roles. Furthermore, *sORF36* and others (*sORF6, 26, 30, 34, 35,* and *40*) overlap with sRNA annotations, suggesting that some of them may represent dual-function regulatory RNAs[78], opening future prospects for functional characterization. Short uORFs, however, are often associated with regulatory functions. In bacteria, tRNA charging levels can reflect nutrient availability, and leader peptides can act as sensors of amino acid scarcity by causing ribosome pausing, thereby regulating downstream gene expression. In *S. aureus*, a leucine-rich leader peptide was described to regulate isoleucine biosynthesis by promoting transcription antitermination under branched-chain amino

acid limitation[53]. Here, we describe a different mechanism, involving leader peptide-dependent regulation of translation initiation. We show that a novel leader peptide, *rbfL*, senses arginine limitation through the presence of rare arginine codons, which are decoded by the low-abundance tRNA$^{Arg(UCU)}$. When arginine is scarce and tRNA charging levels decrease, ribosomes pause on *rbfL* and block the ribosome binding site of the downstream gene *rbf*, preventing its efficient translation initiation. Given that Rbf is an important transcription factor promoting biofilm formation, this regulation likely restricts excess of biofilm formation under arginine-limiting conditions (Fig. 6I). Importantly, pronounced arginine limitation was recently shown to restrict protein synthesis and growth in *S. aureus* biofilms, contributing to antibiotic tolerance[79]. Our findings also align closely with previous studies linking arginine metabolism and biofilm formation, including the role of the conserved sRNA RsaE in promoting biofilm formation[80] and repressing arginine catabolism[81]. Our discovery of arginine-sensitive control of Rbf via *rbfL* further strengthens the connection between arginine metabolism and biofilm formation in *S. aureus*, solidified by our assays that demonstrate increased levels of biofilm in RPMI under exogenous L-Arg supplementation (Fig. S7C). While our study focused on *rbfL*, our data suggest the existence of other regulatory leader peptides, offering exciting directions for future research.

## Methods

All DNA oligonucleotides used in this study are listed in Supplementary Table S1. Information about utilized software, strains, recombinant DNA, reagents, commercial assays and other relevant details are listed in Supplementary Table S3.

### Ribo-seq experiments

The Ribo-seq and Ribo-Ret experiments were performed as previously described[46], largely following the procedure for Ribo-Ret in *E. coli*[50] with several adjustments based on published recommendations[48,67,82,83] and our own optimization for *S. aureus*[46]. For this study, few key steps of the protocol were changed. *S. aureus* HG001 was plated and grown at 37 °C o/n on BHI agar plates. Single colonies were used to inoculate 3 mL of BHI and precultures were grown, shaking at 180 rpm, o/n at 37 °C. The following day, precultures were diluted to an optical density A$_{600}$ (OD) of 0.05 in 50 mL of BHI in 250 mL flasks. Bacterial cultures were grown, shaking at 180 rpm at 37 °C, to an OD of 3.5 and then briefly (5 min) treated with 25 μL of 25 mg/mL Retapamulin (12.5 μg/mL f.c.) for Ribo-Ret. From then, Ret-treated and non-treated samples were processed identically for cell harvest and lysis.

Cultures were subjected to rapid cooling by swirling in an ice-bath for 3 min, before 40 mL of each sample (split into two 50 mL Falcon tubes) were harvested by centrifugation at $2100 \times g$ for 8 min. Another 10 mL of each culture were flash frozen in liquid nitrogen and reserved for total RNA sequencing. For Ribo-seq and Ribo-Ret, cell pellets were resuspended in 550 μL of pre-chilled lysis buffer A (20 mM Tris-HCl pH 8, 50 mM MgCl$_2$, 100 mM NH$_4$Cl, 5 mM CaCl$_2$, 0.4% Triton ×-100, 0.1% NP-40, 100 U/mL DNase 1, 1 mM GMPPNP). Mechanical lysis was performed by two rounds of "fastprep" bead-beating at 6 m/s for 40 s in 2 mL fastprep tubes containing Lysing Matrix B. Subsequently, samples were centrifuged at $16,000 \times g$ for 5 min at 4 °C and supernatants were carefully retrieved. Measurement of absorbance at 260 nm in a Nanodrop spectrophotometer served to adjust nuclease concentration for subsequent digestion by either MNase or RNase 1. For MNase digestion, 750 U of MNase and 50 U of SUPERase-In (does not inhibit MNase) were added per 40 absorbance units (AU). After 1 h shaking at 1400 rpm and 25 °C, the reactions were quenched by addition of 5 mM EGTA (f.c.). For the alternative digestion by RNase 1, 250 U of RNase 1 were added per 40 AU of sample concentration. The digestion was performed for 20 min

shaking at 800 rpm and 37 °C, before it was stopped by addition of 100 U of SUPERase-In. All samples were kept strictly on ice prior to loading onto a 5–50% sucrose gradient for isolation of 70S monosome peaks. RPFs were extracted using acidic hot phenol and size selected on a 15% polyacrylamide- 8 M Urea gel (PAGE). RPFs were gel eluted via passive elution in RNA elution buffer (0.5 M NH$_4$Ac pH 6.5, 1 mM EDTA, 0.1% SDS) o/n at 4 °C shaking at 700 rpm prior to acidic PCI (phenol/chloroform/iso-amyl alcohol, 25/24/1) extraction and ethanol RNA precipitation.

### Library preparation and sequencing analysis

All samples were subjected to ribosomal RNA depletion employing a commercially available rRNA depletion kit according to the manufacturer's protocol. Prior to library preparation, RPFs were de-phosphorylated by the use of Antarctic phosphatase and subsequently phosphorylated with T4 polynucleotide kinase. RPF libraries were then prepared using the NEBNext Small RNA Library Prep Set from Illumina, following the kit's instructions. For the associated total RNA sequencing libraries, a different commercially available kit was employed (NEBNext Ultra II Directional RNA Library Prep Kit). After single-end sequencing on an Illumina NGS instrument, demultiplexed sequencing data in FASTQ format were processed as previously described[46], following published and publicly available workflows[50,82]. To assess ribosome enrichment at start codons globally and to compare resolution of translation initiation peaks between MNase and RNase 1, a global average gene analysis was performed as previously described[50].

### sORF mapping

For the discovery of novel, unannotated sORFs, we followed a similar strategy as described by Meydan et al.[50]. After in silico translation, all potential ORFs with AUG, GUG, or UUG start codons were assigned to Ribo-Ret peaks from duplicate MNase datasets, where 3′ ends of reads mapped within a window of +15 to +21 nt, where e.g., A of AUG designates position +1. Because of higher absolute sequencing coverage (> 25 million vs. approx. 10 million mapped reads per sample), we initially utilized the MNase datasets to search for densities corresponding to novel unannotated sORFs, and later visually inspected all candidate sORFs using the densities of the high-resolution RNase 1 datasets. For this analysis, we considered reads between 32 and 45 nt in length, additionally enriching translation initiation peaks. We then proceeded to filter the dataframe containing all potential unannotated ORFs based on differential rpm expression thresholds (see Fig. S3). We considered all intergenic, unannotated ORFs but also "hypothetical protein" annotations below 100 aa for this analysis and later focused on sORFs below 50 aa in length. Based on SD motif presence ("GGGGG", "GGGG", "GGAGG", "GAGG", "GGAG", or "AGGA") 16–4 nts upstream of start codons, we considered ORFs with RET peak values above 0.1 rpm for visual inspection, but were more stringent with a minimum of 1 rpm in both duplicate experiments for ORFs without predicted SD motif. Because of low, but consistent background levels of Ribo-Ret density within annotated coding sequences globally, presumably due to high levels of pervasive translation, we did not focus our analysis on potential internal initiation sites and excluded them from this mapping approach. Rather, we visually inspected the resulting 944 candidate intergenic sORF initiation sites individually and compared Ribo-Ret densities with the equivalent RNase 1 data. We narrowed down our selection based on mutual expression in the MNase and RNase 1 datasets and constructed a high confidence sORF list. Notably, the majority of final considered densities corresponded to sORFs which had strong SD motif predictions and which had RNase 1 density peaks mapping at appropriate distances to the predicted start codons. After final genome-wide data examination in an integrative genome viewer, 5 additional sORFs were added manually, which had been missed computationally, either due to alternative annotations

(e.g., putative phage protein) or because their densities had mapped diffusely, while other criteria such as strong SD motif and appropriate aligned spacing supported sORF presence. The final curated sORF list is provided in Supplementary Data 1.

## Aligned spacing analysis

The comparatively small genome size of *S. aureus* had allowed us to visually inspect all TISs on a genome browser, individually assessing precise start codon selection, resulting in our latest, improved *S. aureus* HG001 annotation. To study differences in start codon selection and spacing preferences, we calculated the average aligned spacing of *S. aureus* HG001 and the *E. coli* K12 reference genome. The first 20 nt upstream of all annotated start codons were extracted and a motif search was performed considering all instances of "GAGG", "GGAGG", and "GGAG". Exclusively strong, core SD motifs were considered to avoid ambiguous SD-aSD pairing during comparison. For the same reason, sequences with multiple potential SD- motifs were excluded from the analysis, unless the motifs fully overlapped. In few cases of multiple identical sequences extracted, only one instance was considered to avoid confounding effects by duplicate annotations. The nucleotide sequence distance to the start codon was finally counted from the end of the three defined motifs for all upstream sequences identified. To align spacing at the 3′ end of the GGAGG core motif, a one nucleotide spacing count was subtracted in cases of GGAG motifs. The absolute counts of identified motifs were plotted according to their computationally determined aligned spacing (Fig. 2A).

## Extended start site analysis

The 20 nt sequences upstream the start codon, extracted for the aligned spacing analysis, were used to generate a sequence logo, utilizing the Logomaker Python package. Following our biochemical analysis of extended SD-aSD base pairing, these sequences were re-analyzed for SD motif (SD+) and extended SD motif (SD++) presence. To check for the general presence of SD motifs, the first 16 nt (−20 to −5) were searched for any occurrences of GAGG, GGAG, GGGG, GGAGG, AAGG, AGGA, AGGT, or GGGT sequences. We also searched the full 20 nt upstream sequence for occurrences of AGGNAT, AGGNGT, GGGNAT, or GGGNGT, where "N" denotes any nucleotide, to predict the formation of an extended SD-aSD helix. Finally, to analyze the start codon usage, the first three nucleotides of the CDS were extracted and different start codon usage counted.

## Nucleotide sequence conservation analysis

Nucleotide sequence conservation of individual genes has been regularly assessed using curated genomic data publicly available through the SEED database[84]. *S. aureus* genes of interest were first searched for homologs in the database. Their respective coding sequences and upstream nucleotide sequences were extracted, and potential SD motifs and TISs individually assessed. For a broader analysis of sORF conservation, each sORF amino acid as well as nucleotide sequence, encompassing 20 nt of the 5′ UTR and the CDS, were analyzed using tBLASTN and BLASTP for amino acid- and BLASTN for the nucleotide sequences, respectively.

## In vitro transcription

In vitro transcriptions were routinely performed on template PCR products harboring T7 promoter sequence overhangs. Following standard PCR procedures with Phusion DNA polymerase, PCR products were assessed on 1% agarose gels and purified using a commercially available clean-up kit (NucleoSpin Gel and PCR Clean-up, Macherey Nagel). In vitro transcription assays were performed in 400 μL reaction volumes, with a minimum of 1 μg of purified PCR product. Typical reactions contained 4 mM NTPs (final

concentration) each, 0.2 mM Spermidine, 5 mM DTT, 4 μL of RNase inhibitor (RNasin), and 8 μL of home-made T7 RNA polymerase. Reactions were adjusted to 1× reaction conditions making use of a 10× in vitro transcription buffer (0.4 M Tris-HCl pH 8 at 37 °C, 150 mM $MgCl_2$, 0.5 M NaCl). In vitro transcriptions were performed for 3 h at 37 °C. Subsequently, 40 μL of DNase 1 buffer 10× and 10 μL of DNase 1 were added and template PCRs were digested for 1 h at 37 °C. DNase digestion was stopped by addition of 0.5 M EDTA and the resulting RNA was purified by extraction with acidic PCI (phenol/chloroform/isoamyl alcohol, 25/24/1) solution followed by ethanol precipitation. RNA products were subsequently purified on a preparative 6% PAGE and excised from the gel by UV-shadowing. Correct product sizes were gel eluted via passive elution in RNA elution buffer (0.5 M $NH_4Ac$ pH 6.5, 1 mM EDTA, 0.1% SDS) o/n at 4 °C shaking at 700 rpm. A final acidic PCI phenol/chloroform extraction was followed by ethanol precipitation. Purity and quantity were routinely assessed by Nanodrop spectrophotometry.

Notably, the initial PCR reactions for T7 template generation were frequently performed with sequence-modified ssDNA oligo sequences as forward primers to introduce the desired nucleotide changes in the PCR template. This was done, for example, to generate sequence variants for the analysis of the mutant *aur* RBS sequence context, analyzed by toeprinting.

## 30S toeprinting analysis

Toeprinting[85] of *S. aureus* or *E. coli* 30S initiation complexes was performed as previously described[86]. Ternary complexes were formed with in vitro transcribed mRNA, uncharged initiator tRNA^fMet, and purified 30S subunits as follows: 1 pmol of mRNA was hybridized with 2–4 pmol of $^{32}P$-radiolabeled ssDNA toeprinting oligo in a 9 μL reaction volume, briefly denatured for 1 min at 90 °C and refolded by incubation for 1 min on ice and then at room temperature for 5 min. Subsequently, 1 μL of toeprint buffer 10× (100 mM $MgCl_2$, 200 mM Tris−HCl of pH 7.5, 600 mM $NH_4Cl$ and 10 mM DTT) was added. Before use, purified 30S were adjusted to 2 μM concentration in toeprinting buffer 1× and pre-incubated at 37 °C for 15 min. Subsequently, 2 μL of mRNA/oligo mix, 2 μL of 30S, and 1 μL of toeprint buffer 10× were assembled in a final reaction volume of 10 μL. After 10 min incubation at 37 °C, 1 μL of 20 μM *E. coli* deacylated initiator tRNA^fMet was added and samples were incubated for another 5 min at 37 °C. For primer extension, a master mix was prepared so that addition of 2 μL would result in final concentrations of 0.2 mM dNTPs, 1× AMV RT buffer and 1 U/μl of AMV RT. Primer extensions were performed for 30 min at 37 °C. Subsequently, RNA templates were hydrolyzed by addition of 3 μL of 3 M KOH as well as 20 μL of buffer X (50 mM Tris-HCl pH 8.0, 0.5% SDS, and 7.5 mM EDTA) and incubation for 2 min at 90 °C. Subsequently, 6 μL of 3 M acetic acid were added to neutralize the reactions. In the final step, cDNA was recovered by Phenol-Chloroform extraction and ethanol precipitation with addition of sodium-acetate (0.3 M f.c.). cDNA pellets were resuspended in urea loading dye, radioactivity counts were normalized and equal amounts resolved on a denaturing 10% PAGE. Gels were exposed to Fuji X-ray films and developed using an Optimax X-ray film processor. Sequencing lanes were generated as previously described[86].

## 70S toeprinting analysis

For toeprinting of 70S complexes during in vitro translation, a specialized kit was used (*PURExpress delta ribosome kit, NEB*). Toeprinting of in vitro translation reactions was performed as previously described[23]. In brief, following the manufacturer's protocol, a typical reaction contained 2.5 μL of solution A, 0.75 μL of Factor-mix, 1 pmol of gel-purified mRNA template and 0.5 μL of RNase inhibitor (RNasin) in 6.25 μL final reaction volumes. Then, 8 pmol of purified *E. coli* or *S. aureus* 70S ribosomes were supplied. To stall 70S complexes with Retapamulin or Apidaecin-137, their final concentrations were adjusted

to 50 μM in the final reaction volumes. In vitro translations were performed for 30 min at 37 °C prior to addition of 1–2 pmol of $^{32}$P-radiolabeled ssDNA toeprinting oligos. Subsequently, the reactions were supplied with 2 μL of a primer extension mix, providing the four dNTPs at 0.4 mM f.c. each and 2 U of AMV RT per reaction, adjusted to 1× reaction conditions with commercial 5× AMV RT buffer. Primer extensions were performed directly in the toeprint reactions for 30 min at 37 °C. Following cDNA Phenol chloroform extraction, toeprint reactions were resolved as described for 30S toeprinting.

### Reporter plasmid construction

Reporter plasmids were routinely cloned making use of NEBuilder HiFi DNA Assembly to introduce various inserts into a published "prab" vector plasmid backbone[87], carrying constitutively expressed ampicillin and chloramphenicol resistance genes. We had initially introduced a bi-cistronic Renilla- and Firefly reporter cassette, constitutively co-transcribed from a $P_{Blaz}$ promoter sequence variant into a different vector backbone (pCN43)[88]. However, because of the pCN43 encoded erythromycin resistance, we chose to transfer the cassette with its promoter, into the prab vector series with the chloramphenicol resistance. The rationale was to avoid any artificially introduced modifications on the ribosome, which could potentially interfere with the interpretation of our in vivo assays on bacterial start codon selection or ribosome stalling on leader peptides. Inserts for reporter fusions were regularly obtained amplifying PCR products with plasmid-complementary overhangs from the upstream sequences and TISs of genes of interest from HG001 gDNA. These inserts were cloned in-between the bi-cistronic context of the upstream Renilla- and downstream Firefly luciferase sequences, creating translational reporter fusions with the CDS of the latter.

NEBuilder assemblies were generally performed following the manufacturer's protocol, but scaled to a total reaction volume of 4 μL. Both vector backbone and insert PCRs were routinely assessed on 1% agarose gels, and spin-column kit purified following the manufacturer's protocol. Typically, 50 fmol of purified vector backbone PCR were mixed with two to fivefold molar excess of purified insert PCRs for assembly with 2× assembly master-mix. Following 1 h incubation at 50 °C, the reactions were directly transformed into 50 μL of NEB 5-alpha or home-made competent top10 E. coli cells, by standard heat-shock procedures.

For simpler nucleotide sequence changes, either the NEB Q5 site-directed mutagenesis kit or an alternative "quick-change" protocol was employed. Q5 site-directed mutagenesis was performed according to the manufacturer's protocol. For the alternative quick-change protocol, complementary 35–45 nt ssDNA oligos were purchased with matching plasmid sequence context, except for 1–2 nt mismatches at the center of the oligos, designating desired plasmid sequence changes. Between 50 and 200 ng of vector plasmid served as template for amplification by Phusion PCR. Subsequently, 1 μL of DpnI was used to digest template plasmid vector for 1 h at 37 °C, prior to heat-shock transformation.

Plasmids were routinely purified by miniprep utilizing a commercially available kit. All required DNA oligonucleotides were purchased from Integrated DNA Technologies. Every cloned reporter construct was individually verified by Sanger sequencing (Eurofins Genomics).

### S. aureus reporter plasmid transformation

Due to their inherent ease of transformation, S. aureus RN4220 cells were generally employed for in vivo expression of various reporter plasmids. Competent RN4220 cells were prepared and transformed as follows: RN4220 glycerol stock was initially inoculated into 3 mL of BHI for a preculture, grown at 37 °C and shaking at 180 rpm o/n. The following day, a fresh dilution to an OD of 0.05 in 50 mL BHI was grown under equivalent conditions until reaching OD 0.6. At this stage, cells

were split into two 50 mL falcon tubes and pelleted in a pre-chilled centrifuge at 4 °C for 10 min at 2100 × g. Pellets were washed twice by resuspension in 10 mL of 0.5 M sucrose and subsequent centrifugation. After discarding the supernatants, cell pellets were resuspended in 300 μL 0.5 M sucrose each. Then, 100 μL of cells were mixed with 10 μL of plasmid at approximately 500 ng/μL concentration and directly subjected to electroporation. Subsequently, 900 μL of BHI were added, and the cells were transferred to 1.5 mL tubes and left to recover at 37 °C 2 h, shaking at 180 rpm. After recovery, transformed cells were pelleted in a microcentrifuge, resuspended in 100 μL BHI and plated on BHI agar plates containing 10 μg/mL chloramphenicol (Cm) as a plasmid selection marker.

### In vivo dual luciferase reporter assays

Dual luciferase reporter plasmids were routinely transformed into RN4220 competent S. aureus cells, selected via the plasmid-conferred chloramphenicol resistance and stored as glycerol stocks at −80 °C. To perform the dual luciferase assays, these stocks were inoculated into pre-cultures in 3 mL of BHI media containing 10 μg/mL Cm, and grown at 37 °C, shaking at 180 rpm o/n. For each biological replicate, a separate pre-culture was inoculated. The following day, pre-cultures were diluted 1:100 in 5 mL of fresh BHI or RPMI media, containing 10 μg/mL Cm, in 50 mL falcon tubes and grown at 37 °C, shaking at 180 rpm. For rich BHI media, cultures were generally grown for 4 h, reaching mid to late exponential growth phase, whereas cultures in RPMI were grown for up to 7 h. Because of the difference in bacterial growth, the lysis procedure slightly differed. For BHI cultures, 50 μL were directly transferred into 96-well plates, where they were added to 50 μL of PBS solution containing 5 μg of Lysostaphin. Because of the lower optical densities obtained from 7 h of growth in poor RPMI media, 2 mL of these cultures were pelleted in a microcentrifuge and directly resuspended in the equivalent 50 μL of PBS solution containing 5 μg of Lysostaphin, which were then transferred into the 96-well plate instead. S. aureus cell lysis was achieved by subsequent incubation for 30 min at 37 °C. Subsequently, 25 μL of lysate were transferred into an adjacent well, where the dual luciferase assay was performed. First, 25 μL of Promega Dual-Glo firefly luciferase substrate were added and reactions were incubated for 10 min at room temperature. Subsequently, relative light units (RLU) were measured in a Promega GloMax luminometer. According to the manufacturer's protocol, Firefly luciferase was then quenched by the addition of 25 μL of Renilla luciferase substrate reaction mixture (Dual-Glo Stop & Glo) and incubated for another 10 min. Finally, a second measurement of relative light units (RLU) in the luminometer, served to normalize the previously obtained Firefly signal intensities. For all reactions, a negative control was included from BHI or RPMI media assayed in parallel, whose low background signals were subtracted from sample datapoints before normalization.

Because the constitutive $P_{Blaz}$ promoter is also active in E. coli, the initially transformed home-made competent top10 E. coli cells used for cloning could also be used to assay the transformed reporter plasmids in vivo. The respective protocol was largely consistent with the dual luciferase assays performed in S. aureus and only slightly differed in terms of growth conditions and lysis. In detail, plasmid-containing glycerol stocks were inoculated in 3 mL pre-cultures of LB media containing 100 μg/mL ampicillin and grown at 37 °C, shaking at 160 rpm o/n. The pre-cultures were diluted 1:100 in 5 mL fresh LB (Amp) and grown in 50 mL falcon tubes at 37 °C, shaking at 160 rpm for approximately 4–5 h. Next, 2 mL of each culture were pelleted in a microcentrifuge and resuspended in 600 μL of PBS. For lysis, these resuspensions were transferred into 2 mL fastprep tubes containing Lysing Matrix B and subjected to two rounds of FastPrep with the recommended E. coli protocol for mechanical lysis by the MP Biomedicals FastPrep system. Subsequently, the tubes were spun down at maximum speed in a microcentrifuge and 25 μL were directly

transferred into 96-well plates for dual luciferase assay as described for *S. aureus* cell lysates.

## *S. aureus* ribosome purification

*S. aureus* 70S ribosomes were purified as previously described[70]. Briefly, fresh o/n precultures were diluted to an $OD_{600}$ of 0.05 in two liters of BHI media and grown at 37 °C shaking at 180 rpm to an $OD_{600}$ of 1.0, reaching logarithmic growth phase. Cells were harvested by centrifugation at $2100 \times g$ at 4 °C and washed twice with cold buffer A (20 mM HEPES-KOH pH 7.5, 100 mM $NH_4Cl$, 21 mM $Mg(CH_3COO)_2$, 1 mM DTT). Approximately 1.5 g of bacterial cell pellet was resuspended in 10 mL of buffer A, supplemented with 0.75 mg of lysostaphin, 150 U of DNase 1, 24 μL of 0.5 M DTT as well as 50 μL of Protease Inhibitor ×100 and then incubated at 37 °C for 45 min. Cell lysate was cleared by centrifugation at $30,000 \times g$ for 90 min. The supernatant was kept and supplemented with PEG 20,000 until a concentration of 2.8% w/v was reached and subsequently centrifuged at $20,000 \times g$ for 5 min. The supernatant was recovered again and further PEG 20,000 added to adjust the percentage to 4.2% w/v before centrifugation at $20,000 \times g$ for 10 min. The resulting pellet was resuspended in 35 mL of buffer A and carefully layered on a 25 mL sucrose cushion (10 mM HEPES-KOH pH 7.5, 500 mM KCl, 25 mM $Mg(CH_3COO)_2$, 1.1 M sucrose, 0.5 mM EDTA, 1 mM DTT) before centrifugation at $146,900 \times g$ (40,000 rpm) for 16 h at 4 °C in a T70i rotor (Beckman Coulter). The ribosome pellet was washed twice with 5 mL of buffer E (10 mM HEPES-KOH pH 7.5, 100 mM KCl, 10 mM $Mg(CH_3COO)_2$, 0.5 mM EDTA, 1 mM DTT) before resuspension in 1.5 mL of buffer E. The resuspensions were evaluated by Nanodrop spectrophotometry and approximately 200 AU layered on top of a 7–30% sucrose gradient. Gradients were centrifuged at $49,900 \times g$ (17,100 rpm) for 16 h 40 min at 4 °C in an SW32 rotor (Beckman Coulter). Sucrose gradient tubes were fractionated with a Piston fractionator (Biocomp). Samples corresponding to 70S ribosomes were collected, pooled and the magnesium concentration was adjusted to 25 mM. Subsequently, a final PEG precipitation was performed by addition of PEG 20,000 up to 4.5% w/v and centrifugation at $20,000 \times g$ for 12 min. Ribosomal pellets were resuspended in buffer G (10 mM HEPES-KOH pH 7.5, 50 mM KCl, 10 mM $NH_4Cl$, 10 mM $Mg(CH_3COO)_2$, 1 mM DTT) and flash frozen in liquid nitrogen until use for biochemical assays or structure determination by cryo-EM.

## Complex assembly and cryo-EM grid preparation

*S. aureus* 70S ribosomes were adjusted to 7.5 μM in toeprint buffer 1× (10 mM $MgCl_2$, 20 mM Tris–HCl of pH 7.5, 60 mM $NH_4Cl$, and 1 mM DTT) and activated for 15 min at 37 °C. Subsequently, 1 μL of activated 70S was mixed with 8 μL of distilled water and 1 μL of 10× toeprint buffer $\Delta Mg^{2+}$ (200 mM Tris–HCl of pH 7.5, 600 mM $NH_4Cl$, and 10 mM DTT), adjusting the magnesium concentration to 1 mM for a "70S breathing" step. 70S at 1 mM $Mg^{2+}$ were incubated for 15 min at 37 °C prior to complex assembly. In parallel, in vitro transcribed and purified *aur* mRNA was unfolded (90 °C, 1 min), snap-cooled on ice (1 min), and refolded in toeprint buffer 1× at room temperature (5 min). Complexes were assembled by mixing 70S ribosomes, *aur* mRNA, and deacylated initiator tRNA$^{fMet}$ at a 1:7:7 molar ratio in 1× toeprint buffer ($\Delta Mg^{2+}$) and raising $Mg^{2+}$ concentration to 10 mM for 15 min at 37 °C, followed by a further increase to 15 mM and incubation for 15 min at 37 °C. Final assembly concentrations were 300 nM 70S, 2.1 μM *aur* mRNA, and 2.1 μM itRNA$^{fMet}$.

For cryo-EM, 3 μL of the *S. aureus* 70S initiation complex were applied to glow-discharged Quantifoil R2/2 300-mesh carbon-coated copper grids, blotted, and plunge-frozen using a Vitrobot IV (Thermo Fisher Scientific) at 95% humidity and 10 °C. Data were collected on a Titan Krios G4 equipped with a Selectris X energy filter and a Falcon 4i detector as electron-event representation movies (nominal magnification of $165,000 \times g$; pixel size 0.729; total fluence 30 e⁻/Å²).

## Image processing and atomic model building

Movies (11,450) were processed using the software CryoSPARC (v4.7)[92]. Patch Motion Correction and CTF estimation were followed by particle picking, yielding 1,931,605 particles extracted and coarsened 3×. After 2D classification, 1,373,556 particles were used for ab initio reconstruction and homogeneous refinement. To sort different conformations, 3D classification was used to identify 5 main classes: (1) rotated 70S with E-site tRNA (516,117 particles), (2) non-rotated 70S with E- and P-site tRNAs (383,073 particles), (3) non-rotated 70S with E- and partially occupied P-site tRNAs (242,930 particles), (4) 50S (170,566 particles) and (5) junk/unassigned (60,870 particles).

Particles from classes (2) and (3) were subjected to focused classification and refinement[93–98] using a mask covering the density corresponding to the P-site tRNA and the SD-aSD helix. A final set of 316,031 particles was re-extracted at original pixel size. Angular distributions of the particles were assessed with the program VUE[99]. The extracted particles were subjected to non-uniform refinement with defocus and global CTF refinement, resulting in a 2.3 Å map as estimated from Fourier Shell Correlation[89–91].

The structure of the *S. aureus* 70S initiation complex (PDB ID: 6YEF[100]) was rigid-body fitted in UCSF Chimera software. Atomic model building and inspection were carried out in the software Coot v0.9.8.95, including placement of the mRNA segment AGGAGGAAUGAAAUGUG and the 16S rRNA segment GGAUCACCUCCUUU using Coot. The model was iteratively refined by real-space refinement in the software Phenix v1.21 with manual adjustments in Coot, until convergence.

Cryo-EM data collection- and processing statistics are further detailed in Supplementary Table S2. The workflow of the Cryo-EM data processing is further outlined in Supplementary Fig. S8.

## *S. aureus* biofilm formation assay

Overnight precultures were prepared by inoculating *S. aureus* HG001 from glycerol stocks into 3 mL BHI and incubating at 37 °C, shaking at 160 rpm. Cultures were diluted 1/100 into RPMI 1640 (Merck R7509) with or without L-Arg (4 g/L). Aliquots (200 μL) were distributed into wells of a 96-well polystyrene plate (Thermo Fisher); peripheral rows were filled with 200 μL of RPMI to minimize edge effects. For each condition, 24 technical replicates were performed per plate. Plates were incubated statically for 48 h at 37 °C. Supernatants were then carefully transferred to a fresh 96-well plate to quantify planktonic growth (OD 600 nm; Multiskan Skyhigh, Thermo Fisher).

Attached biofilms were fixed with 200 μL of methanol for 10 min at room temperature, methanol carefully removed, and wells air-dried for 5 min. Biofilms were stained with 200 μL 0.1% (w/v) crystal violet (Sigma) for 1 h at room temperature, washed gently with 200 μL distilled water, air-dried for 5 min, and resuspended with 200 μL of 33% (v/v) acetic acid. Crystal violet signal was quantified at OD 595 nm using the same plate reader.

## Statistics and reproducibility

Toeprint experiments in Figs. 2c, 3e, 5b–d, and 6b provide qualitative information about the position of ribosome recruitment on the respective mRNAs. No quantitative information for statistics was derived from them. All representative toeprint experiments shown were performed three times.

## Reporting summary

Further information on research design is available in the Nature Portfolio Reporting Summary linked to this article.

## Data availability

Ribosome profiling (Ribo-seq) and associated total RNA sequencing (RNA-seq) data applied in this study are available in the Gene

Expression Omnibus (GEO) database under accession numbers GSE299221 and GSE299222, respectively. The cryo-EM map and the atomic model for the *S. aureus* 70S ribosome complex have been deposited in the Electron Microscopy Data Bank (EMDB) and in the Protein Data Bank (PDB) with accession codes EMD-55526 and PDB: 9T4R, respectively. The raw reporter expression data and all raw/uncropped toeprint gels are provided in the Source Data. Source data are provided via figshare [https://doi.org/10.6084/m9.figshare.29480363].

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

## Acknowledgements

We are grateful to Dr. Jose Refugio Jaramillo (IBMC, Strasbourg) for his technical assistance and continuous support throughout various aspects of the project. We thank Prof. Alexander Mankin (University of Illinois, Chicago) for his ongoing insights into the methodology, his close engagement with the progression of our results, and his valuable advices. We also thank Prof. Wolfgang R. Hess (University of Freiburg) for helpful discussions on the overall project. We acknowledge Isabelle Caldelari and David Lalaouna for their fruitful discussions. We thank Sasha Ballet, Tan Dat Truong & Léo Fréchin for IT support, and Alexandre Durand & Nils Maréchal for cryo-EM support at the integrated structural biology platform of the CBI. This work was supported by the French National Research Agency ANR (SaRNAmod: ANR-21-CE12-0030-01, SatRNAsPG ANR-24-CE11-7652 to SM, IntRNAReg ANR-23-CE12-0041-01 to PR & SM). This work of the Interdisciplinary Thematic Institute IMCBio+, as part of the ITI 2021-2028 program of the University of Strasbourg, CNRS, and Inserm, was supported by IdEx Unistra (ANR-10-IDEX-0002), and by SFRI-STRAT'US project (ANR 20-SFRI-0012) and EUR IMCBio (ANR-17-EURE-0023) under the framework of the French Investments for the Future Program. The electron microscope facility at the CBI/IGBMC was supported by the Region Grand Est, FEDER, the French Infrastructure for Integrated Structural Biology (FRISBI) ANR-10-INBS-0005/France 2030 program, EquipEx+ France-Cryo-EM (ANR-21-ESRE-0046), and Instruct-ERIC.

## Author contributions

Project conceptualization, S.M., M.P.K., and P.R.; methodology, M.P.K., S.M., R.B.C., and B.P.K.; investigation, M.P.K., R.B.C., M.K., and M.D.E.; analysis, M.P.K. and B.C.W.M.; cryo-EM data collection, C.B. and O.V.L.; cryo-EM analysis, R.B.C.; visualization, M.P.K., R.B.C. and S.M.; funding acquisition, S.M., P.R., and B.P.K.; supervision, S.M., P.R., and B.P.K.; writing–original draft, M.P.K., S.M., and P.R.; writing–review & editing, M.P.K., S.M., P.R., and B.P.K., with inputs from everyone.

## Competing interests

The authors declare no competing interests.
