## [Transparent Peer Review file · Nature Communications]

Extended Shine-Dalgarno motifs govern translation initiation in *Staphylococcus aureus*

Corresponding Author: Dr Stefano Marzi

Version 0:

Reviewer comments:

Reviewer #1

(Remarks to the Author)

Translation initiation is a critical step during protein synthesis since it determines the reading frame of the open reading frame (ORF) to be translated, as well as representing a rate limiting step governing the efficiency of translation. Despite decades of research, a complete understanding of this process is lacking in the well-studied Gram-negative bacterium *E. coli*, let alone in many other bacteria, including Gram-positive pathogenic bacteria, such as *Staphylococcus aureus*. In the study of Kohl et al, the authors employ Ribo-seq and Ribo-RET using RNase I to obtain high quality information of translation start sites in *S. aureus* revealing numerous new unannotated small ORFs. The authors reveal that *Staphylococcus aureus* uses an extended Shine-Dalgarno interaction to modulate start site selection and demonstrate that this extended Shine-Dalgarno interaction generates species-specificity with regard to *E. coli*. The authors also demonstrate that one of the newly identified ORFs acts as an upstream ORF (uORF) to regulate translation of a downstream ORF involved in biofilm formation. Collectively, the manuscript provides, on one-hand, an excellent resource for those working with *Staphylococcus aureus*, and, on the other hand, novel fundamental insight into the regulation of translation initiation that will be of general interest to those working in the fields of translation control.

The strength of the manuscript is definitely the high quality of the Ribo-seq/Ribo-RET data, leading to identification of novel ORFs. The weakness of the manuscript is that the authors have followed-up on two distinct observations that are not quite taken to their full conclusion, one related to the extended Shine-Dalgarno interaction and the other related to the regulation of the *rbf* gene via the *rbfL* leader peptide. A few additional clarifications would significantly improve the manuscript.

1. While the application of RNase 1 to obtain high resolution Ribo-seq data is in itself not novel, having been previously used for other bacteria, such as *Listeria* and *Salmonella*, the results of the Ribo-seq and RiboRET are completely novel for *Staphylococcus*. Indeed, the authors identify 46 unannotated small ORFs, of which the initiation sites were validated using in vitro toeprinting assays. However, unless I missed it, it is not made clear whether the novel small ORFs are conserved at all in other bacteria or whether they are only present in *Staphylococcus* or related species?
2. The authors identify an extended Shine-Dalgarno interaction in *Staphylococcus* that provides some species-specific start codon selection compared to *E. coli*. This is an exciting finding however some clarification is required for the mechanistic model that is proposed, particularly with respect to the cryo-EM structure that is presented. For a start, it is surprisingly that the authors conduct their biochemical analysis using the *aur* mRNA (Fig. 3c and 3d), yet the structure is generated with another mRNA, namely, the *spa* mRNA? Therefore, it is hard to correlate the biochemical mutagenesis with the cryo-EM structure. For example, the *spa* mRNA contains a GGGGGU such that the U can basepair with the A of the aSD but there is no complementary base to the C of the aSD so it remains unbasepaired. However, the *aur* mRNA has GGAGGA such that the A cannot basepair with the A of the aSD. For *cvf1*, it is unclear to me why the C in the aSD is still unpaired and does not basepair with the G in the SD sequence as shown for *ccpN*? In fact, what is the basis for the basepairing schemes that are used in this study?
3. Based on the cryo-EM structure, the authors propose that the extended SD-helix results from two additional A-U basepairs with an unpaired C in the aSD and with a 4nt spacing from the SD to the UUG start codon. This may well be true, however, the way the complex has been formed raises many concerns about the relevance of this structure...One would have expected that the authors would have added the mRNA to the 70S ribosomes in the presence of fMet-tRNA, however, this appears not to be the case. Rather 70S ribosomes were incubated for 10mins at 37deg with 1mM magnesium (to split

them?) and then mRNA and deacylated tRNA^{fMet} (why not fMet-tRNA?)...and then non-aminoacylated tRNA^{Lys} – why was this added at all? And then subsequently the magnesium was increased to 15mM. It is unclear to me what to expect under these conditions...do all 70S ribosomes get programmed with deacylated tRNA^{fMet} in the P-site and tRNA^{Lys} in the A-site, or can they also be programmed with tRNA^{Lys} in the P-site and tRNA^{fMet} in the E-site? The methods on page 25 suggest that the major states are “ (1) non-rotated with E-site tRNA, (2) rotated 70S with A-site tRNA, (3) rotated 70S with A- and P-site tRNAs...” however, the actual number of particles needs to be detailed here. Also it is hard to envisage rotated ribosomes with canonical tRNAs...presumably they are hybrid given that deacylated tRNAs were used? Also it is hard to imagine a state with only A-tRNA and no P-tRNA? In this regard, the text appears to differ from the Sup Fig 6 which suggests that the classes that were merged contained P- and A-tRNAs, and P- and E-tRNAs although here it is also not mentioned whether they are rotated or not? Since the accompanying map appears to have density for A-, P- and E-tRNAs, one can presume that these later (non-rotated) states were merged. This raises the question whether the authors could be sure that the P-tRNA in both cases is actually the initiator tRNA? This is important since if the P-tRNA in one case was the tRNA^{Lys} with the initiator tRNA in the E-site, then the authors have merged two states with different linker lengths to the extended Shine-Dalgarno helix. It would seem appropriate to convince the reader that these substates are the same before merging. The cryo-EM map provided has been highly filtered and very tightly masked – it would seem appropriate for the authors to provide the unsharpened 3d refine and post-processed maps so one can assess the correlation between the map and models before the sharpening process has occurred.

4. Overall, it is unclear why the authors also only collected 1105 movies, yielding only 150K particles and limiting themselves to lower resolution – especially given that it is important for the study to understand exactly which tRNAs are bound in which positions so as to have the register and resolution to know how the SD and aSD are basepaired. I am not convinced this is possible with the resolution that they report.

5. The additional base-pairing formed in *Staphylococcus* is proposed to be possible because of the lack of S21, and that the presence of S21 in *E. coli* prevents this extended helix forming. This is not taken to its conclusion by demonstrating that *E. coli* 70S lacking S21 can also form extended helices and therefore initiate translation as observed in *Staphylococcus*.

6. While the authors convincingly show that rbfL is involved in regulating the expression of the downstream rbf gene, some questions still remain. The authors propose that the presence of rare AGA and AGG codons in the rbfL uORF causes stalling under poor nutrient conditions. This is supported by experiments in Figure 6C-G. It is unclear to me why the authors didn't simply substitute the rare arginine codons for the others ones (CGU, CGC, CGA, CGG) but rather mutate one of them to a glycine codon? It is also unclear to me why addition of extra L-Arg should help translation through rare codons - or are the authors proposing that there is also some charging problem due to Arg limitation? The Y-axis of graphs C-G are all different making it hard to compare between conditions – maybe one should not but still it would be better to have the same axis values. The rescue by 10X L-Arg is better than by expression of the tRNA^{Arg}(UCU) – this doesn't appear to really support the rare Arg codon theory?

7. Finally, the overall model for rbfL regulation of rbf expression is that stalling near the stop codon of rbfL prevents access of 30S subunits to the SD and start codon of the downstream rbf gene, however, the authors do not demonstrate that ribosomes stall, let alone where they stall in the rbfL ORF. This would seem like an important experiment to perform before one can make such a claim as shown in Figure 6H.

Some additional minor points:

1. Abstract line 26: What is striking about it compared to the many known examples? This is quite subjective, perhaps remove the word striking?
2. Line 51: Perhaps also include citation to PMID: 17355865 from Yokoyama group that showed SD interaction on 30S subunit, rather than 70S, at higher resolution than reference 13 also in the same year (2007)
3. Line 75: I thought reference 46 showed that they could obtain high resolution despite using MNase?

Reviewer #2

(Remarks to the Author)

In bacteria, a major cis-acting sequence element influencing translation initiation is the Shine-Dalgarno (SD) sequence, which facilitates ribosome binding upstream of start codons. Building on this concept, this manuscript presents a high-resolution analysis of translation initiation in *Staphylococcus aureus*, a clinically relevant pathogen. Using RNase I-based ribosome profiling in combination with Retapamulin treatment (Ribo-Ret), the authors accurately map translation start sites across the *S. aureus* transcriptome and also monitor the usage of rare, unconventional start codons. In addition, the authors identify numerous previously unannotated small open reading frames (ORFs), including regulatory upstream ORFs (uORFs).

A key finding is that *S. aureus* initiation involves extended Shine-Dalgarno motifs, which form strong base-pairing interactions with the anti-SD region of the 16S rRNA. Notably, the study identifies two non-canonical start codons, AUU and AUA, whose recognition depends on these extended SD sequences. The authors further investigate a biologically significant case of uORF-mediated regulation: translation of a small leader peptide modulates expression of rbf, a transcription factor essential for biofilm formation. This regulatory mechanism is conditionally governed by codon usage bias and arginine availability.

Overall, this study reveals key differences from canonical translation initiation mechanisms described in *E. coli*, highlighting unique aspects of translational regulation in *S. aureus*. In this regard, the work constitutes a significant contribution to the field and deserves recognition, even if certain ambiguities and limitations remain.

Major comments:

1- Line 52–53: The sentence discussing the S1 ribosomal protein is very vague. The specific role or relevance of this protein in the context of the study should be clarified.

2- Line 84–86 as well as throughout the manuscript: This analysis raises the question of how different the ribosomes of *E. coli* and *S. aureus* are, also with regard to the structure obtained through cryo-EM (Figure 3B) It would be helpful to briefly describe them or clarify whether such differences were systematically analyzed. Why does an extended basepairing between SD and anti-SD necessitate a longer linker sequence upstream of the start codon in *S. aureus* (2A)?

3- Lines 114–121: It appears that the authors refer to the “no drug” condition as a Ribo-seq dataset. However, capturing ribosome-protected fragments typically involves an elongation-stalling antibiotic to arrest translating ribosomes. Could the authors clarify how ribosome footprints were obtained in the absence of such treatment?

4- Line 116: The observation of larger ribosome footprints is indeed intriguing. While the authors propose that this may be due to extended SD–anti-SD interactions, it is noted that only 28% of mRNAs possess such extended SD motifs (Figure 3F). This proportion seems insufficient to account for the dominant larger peak observed in Figure 1D. Is this distribution also seen in comparable experiments in other species, and which other factors may contribute to this footprint size distribution?

5- Figure 1E (and similar graphs): It would be more intuitive to present the sequence in the graph in the 5'→3' orientation to facilitate sequence comparisons.

6- Figure 1E: It is unclear why the upper panel displays the RNase I control and RNase I + Retapamulin datasets, while the zoom-in panel below shows the MNase + Retapamulin and RNase I control datasets. For clarity and consistency, it could be an option to show all datasets? Is the size bar in the upper panel (ctrl) correctly labelled, as it differs significantly from the others?

7- Figure 2D: The luciferase assay suggests that *S. aureus* 70S ribosomes prefer both extended SD sequences and GUG as a start codon. It would be interesting to test whether moving the GUG start codon closer to the SD (e.g., in a GUGAUG context) changes the result in the toeprinting assay. Some studies (e.g., Yamamoto et al., 2016, PNAS) suggest ribosomes may favor the first start codon from the 5' end. Is initiation mainly driven by the SD interaction, or also by start codon identity and in-frame position?

8- Figure 3D: A brief interpretation of the *E. coli* results with regard to the impact of the SD sequence would be helpful.

9- Figure 3F–G: The figures show the percentage of mRNAs with extended SD sequences and start codon usage. Could the authors also indicate which start codons most often co-occur with extended SD motifs?

10- Figure 3F: Do genes with no or short SD sequences have distinct features? For example, could reduced secondary structure around the start codon promote efficient SD-independent initiation? A brief analysis or comment on this would be helpful.

11- Figure 5: The authors report 17 uORFs, of which 8 were tested by toeprinting and 5 by luciferase assay. Could the authors clarify the criteria used to select uORFs for each experiment? This would help readers understand the rationale behind the experimental choices.

12- Figure 6C–E: Cross-comparison between panels is difficult due to the different y-axis scales. Despite that, could the authors clarify why the *rbf*-uORF mutant behaves differently in RPMI with or without 10× L-Arg? Also, wouldn't we expect *rbfL* expression to be similar in RPMI + 10× L-Arg and RPMI + tRNAArg?

13- Figure 6G: what is the effect of the amino acid substitution on the expression of *rbfL*?

14- Figure 6H: The authors propose that the 30S cannot initiate due to a stalled 70S on *rbfL*. However, previous studies have demonstrated “70S scanning initiation,” where post-termination 70S ribosomes can scan downstream for an SD sequence. Could the authors comment on this alternative mechanism and whether it may apply in this context?

15- Figure 6: The authors propose that ribosome stalling on a small uORF reduces translation of the downstream gene *rbf*, which is involved in biofilm formation. Has this hypothesis been tested by assessing biofilm formation in nutrient-poor media?

16- The proposed ribosome stalling at the uORF is intriguing. Have the authors considered Ribo-seq in poor media to explore this? Not essential, but it could offer useful insights.

17- Figure S3: The start codon from the old annotation is not marked. Could the authors zoom out or adjust the figure to indicated the position of the old start codon for comparison?

18- Figure S5: In *E. coli*, *infC* initiates from a non-canonical start codon (AUU). Does it also have an extended SD sequence, or is this feature unique to *Staphylococcus*? Similarly, for the other conserved genes with non-canonical start codons, does the link to extended SD motifs occur only in *Staphylococcus*, or is it more broadly conserved?

Minor comments:

- 1- Fig. 1G: this representation suggests that STAR sORFs may have a “negative length”.
- 2- Line 33: Consider introducing the abbreviation "S. aureus" when first mentioning *Staphylococcus aureus*.
- 3- The usage of "*Staphylococcus aureus*" and "S. aureus" is inconsistent throughout the text. Please standardize the abbreviation after its first introduction
- 4- Line 230: reference to Fig. 3 A, D should be Fig. 3 C, D

Reviewer #3

(Remarks to the Author)

Reviewer #4

(Remarks to the Author)

The paper of Kohl et al. explores the principles of translation initiation and start codon selection by the ribosome in *Staphylococcus aureus*. To this end, the authors utilized Ribo-seq analysis of the cells treated with retapamulin, a drug that selectively stalls translating ribosomes at start codons (Ribo-Ret). The Ribo-Ret was previously used to map start sites in *E. coli*. However, in this case, the digestion of the polysome fraction with MNase, a nuclease with a strong sequence bias, limits precise mapping of ribosome-protected fragments and therefore the identification of start sites (especially in cases where several potential start sites are close to each other). In this paper, the authors used RNase I instead of MNase, which allowed them to significantly improve the quality of mapping of initiation sites in *S. aureus*. Further analysis of Ribo-Ret data, combined with structural studies, showed that this bacterium relies on a specific SD-aSD interaction to select the start codon, which is longer than the corresponding interaction in *E. coli*. This conclusion is supported by an elegant in vitro toe-print analysis. The authors further demonstrate that such extended SD-aSD interaction can be utilized by *S. aureus* (and likely other Firmicutes) for uORF-mediated translational control of downstream gene expression, including those that regulate biofilm formation in this human pathogen.

This is an exciting and illuminating study that significantly extends our understanding of translation initiation in bacteria. The experiments are very 'clean', convincing, and properly controlled. The paper is well-written and an easy read. The study represents novel findings and is appropriate for publication.

Version 1:

Reviewer comments:

Reviewer #1

(Remarks to the Author)

The authors have made a tremendous effort to address the reviewer comments including an additional cryo-EM structure on the aur mRNA that complements nicely the biochemistry. Additional experiments have also been performed addressing the mechanism of action of the RbfL leader peptide. I congratulate the authors on an exceptional piece of work, which I am sure will be of high interest and importance to the translation community. I therefore fully support rapid publication of this article in Nature Communications.

Reviewer #2

(Remarks to the Author)

We thank the authors for the comprehensive replies to our comments. We are satisfied that all criticisms and suggestions for improvements have been addressed, and that the missing details have been clarified.

Reviewer #3

(Remarks to the Author)

Reviewer #1 (Remarks to the Author)

Translation initiation is a critical step during protein synthesis since it determines the reading frame of the open reading frame (ORF) to be translated, as well as representing a rate limiting step governing the efficiency of translation. Despite decades of research, a complete understanding of this process is lacking in the well-studied Gram-negative bacterium *E. coli*, let alone in many other bacteria, including Gram-positive pathogenic bacteria, such as *Staphylococcus aureus*. In the study of Kohl et al, the authors employ Ribo-seq and Ribo-RET using RNase I to obtain high quality information of translation start sites in *S. aureus* revealing numerous new unannotated small ORFs. The authors reveal that *Staphylococcus aureus* uses an **extended Shine-Dalgarno interaction** to modulate start site selection and demonstrate that this extended Shine-Dalgarno interaction generates species-specificity with regard to *E. coli*. The authors also demonstrate that one of the newly identified ORFs acts as an upstream ORF (uORF) to regulate translation of a downstream ORF involved in biofilm formation. Collectively, the manuscript provides, on one-hand, an excellent resource for those working with *Staphylococcus aureus*, and, on the other hand, novel fundamental insight into the regulation of translation initiation that will be of general interest to those working in the fields of translation control.

We thank the reviewer for this positive assessment and for recognizing both the significance and broad relevance of our work.

The strength of the manuscript is definitely the high quality of the Ribo-seq/Ribo-RET data, leading to identification of novel ORFs. The weakness of the manuscript is that the authors have followed-up on two distinct observations that are not quite taken to their full conclusion, one related to the extended Shine-Dalgarno interaction and the other related to the regulation of the *rbf* gene via the *rbfL* leader peptide. A few additional clarifications would significantly improve the manuscript.

We thank the reviewer for highlighting the two central findings of our study, both derived from the high-resolution Ribo-seq and Ribo-Ret analyses. These datasets allowed us to precisely map all translation initiation events in *S. aureus*, resolve misannotated start sites, and identify numerous new small proteins, allowing a major improvement of genome annotation. This comprehensive view of initiation sites indeed enabled us to explore two distinct but complementary aspects, which govern protein synthesis in this major human pathogen: (i) the mechanistic basis of translation initiation, focusing on the extended SD-aSD interactions and their biochemical significance, and (ii) a mechanism of regulation of translation initiation involving the functional characterization of a novel small uORF encoding the leader peptide *rbfL*. According to the reviewer's suggestions, we bring more clarifications in order to render our message clearer and more cohesive.

1. While the application of RNase 1 to obtain high resolution Ribo-seq data is in itself not novel, having been previously used for other bacteria, such as *Listeria* and *Salmonella*, the results of the Ribo-seq and Ribo-RET are completely novel for *Staphylococcus*. Indeed, the authors identify 46 unannotated small ORFs, of which the initiation sites were validated using in vitro toeprinting assays. However, unless I missed it, it is not made clear whether the novel small ORFs are conserved at all in other bacteria or whether they are only present in *Staphylococcus* or related species?

We thank the reviewer for recognizing the novelty and quality of our high resolution Ribo-seq and Ribo-Ret datasets for *S. aureus*. While RNase 1 has been previously applied in *Listeria* and *Salmonella* (Bryant et al., 2023), our study provides the first genome-wide, single-nucleotide resolution mapping of translation initiation sites in bacteria, achieved through the combined use of RNase 1 and the initiation inhibitor Retapamulin. Importantly, as shown in Figure 1 of the manuscript, our optimized lysis and footprinting conditions yielded exceptionally high read length precision and frame assignment quality.

Regarding sORF conservation, we initially focused our analysis on putative regulatory uORFs (manuscript Figure 5A) and used a targeted approach, identifying homologs of downstream genes in related species via 'The Seed' database and analyzing corresponding upstream regions. This strategy allowed us to detect conserved uORFs in several *Staphylococcus* species, such as the highly conserved uORF sORF18 (*gehAL*; manuscript Figure 5D and the Figure R1, below).

Figure R1: Conservation of *gehAL* (sORF18). Exemplary uORF *gehAL* showing high degree of conservation in staphylococcal species, identified by analyzing *gehA* homologous using 'The Seed' database. (A) The genomic locus and Ribo-RET profiles of *gehA* and *gehAL*, with the translation initiation sites (TIS) indicated. (B) Sequence conservation of the *gehAL* uORF, including the TIS region and predicted peptide.

Following the reviewer's suggestion, we have now performed a comprehensive conservation analysis for all identified sORFs using BLASTN, tBLASTN, and BLASTP. The results, now included in Supplementary table S1, revealed extensive conservation for multiple sORFs, which is mostly restricted to Staphylococcaceae, including sORFs 15, 16, 18, 20, 24, 28, 29, 30, 34, 43, and 45. These findings have been incorporated into the Results section: '*Finally, using a combination of BLASTN, tBLASTN, BLASTP and targeted database searches, we identified high conservation for several sORFs, in particular for sORFs 15, 16, 18, 20, 24, 28, 29, 30, 34, 43, and 45, mostly restricted to Staphylococcaceae (Table S1).*'.

Reference:

Bryant OJ, Lastovka F, Powell J, Chung BYW. The distinct translational landscapes of gram-negative Salmonella and gram-positive Listeria. Nature Communications. 2023;14(1):8167.

2. The authors identify an extended Shine-Dalgarno interaction in *Staphylococcus* that provides some species-specific start codon selection compared to *E. coli*. This is an exciting finding however some clarification is required for the mechanistic model that is proposed, particularly with respect to the cryo-EM structure that is presented. For a start, it is surprisingly that the authors conduct their biochemical analysis using the *aur* mRNA (Fig. 3c and 3d), yet the structure is generated with another mRNA, namely, the *spa* mRNA? Therefore, it is hard to correlate the biochemical mutagenesis with the cryo-EM structure.

We thank the reviewer for this important comment. We agree that in the original submission, direct correlation between the biochemical analyses performed on *aur* mRNA and the cryo-EM structure obtained with *spa* mRNA was limited, as these mRNAs have slightly different spacer and RBS sequence contexts. The choice of *aur* mRNA was guided by its natural AUGUG start-site overlap and accurate GUG start codon selection *in vivo*, as revealed by our Ribo-RET data, making it an ideal candidate for biochemical validation.

To address this critical point, we have now determined a new cryo-EM structure of the *S. aureus* 70S initiation complex programmed with *aur* mRNA, the same mRNA used for biochemical analyses. This new structure confirms the presence of an extended SD-aSD interaction centered around the predicted AU dinucleotide pair, consistent with our mutagenesis data, and provides additional molecular details that strengthen our mechanistic model. Importantly, the positioning of the SD-aSD is very similar in both *spa* and *aur* ribosomal complexes. The Results section and Figure 3A have been modified accordingly to reflect the new data.

For example, the *spa* mRNA contains a GGGGGU such that the U can basepair with the A of the aSD but there is no complementary base to the C of the aSD so it remains unbasepaired. However, the *aur* mRNA has GGAGGA such that the A cannot basepair with the A of the aSD.

We thank the reviewer for this insightful observation. Our new cryo-EM structure clarifies how extended SD-aSD interactions are accommodated in the *aur* mRNA, despite the sequence differences relative to *spa* mRNA. It now shows that direct base-pairing between the uridine following the canonical SD-motif (GGGGGU on *spa* mRNA versus GGAGGA in *aur*) and A1546 of the 16S is not required for extended SD-aSD formation. In the *aur* complex, A1546 of the aSD is flipped out, while the respective adenosine on the mRNA remains within a helical conformation. This configuration is stabilized by stacking interactions with neighboring bases and a hydrogen bond with C1545 of the aSD. These observations demonstrate the structural flexibility of the 16S rRNA 3' end in accommodating extended SD-aSD helices.

Due to the limitations of our previous structure with *spa* mRNAs (resolution and presence of a second tRNA in the A-site, see below) and to keep this work focused we decided to exclusively discuss the *aur* structure in the revised paper.

For *cvfc1*, it is unclear to me why the C in the aSD is still unpaired and does not basepair with the G in the SD sequence as shown for *ccpN*? In fact, what is the basis for the basepairing schemes that are used in this study?

We thank the reviewer for raising the question of whether specific rules can be used to identify extended base-pairing schemes for any mRNA. Indeed, the precise nature of extended SD-aSD interactions cannot yet be fully predicted for every mRNA (e.g., *cvfC1* or *ccpN*), and we have therefore revised all figures and text to refer to such alignments as 'potential extended SD-aSD' unless directly supported by structural data.

Our rationale for the indicated base-pairing schemes is based on both structural and biochemical evidence (toeprinting etc.). Based on biochemical data for *aur* mRNA, we proposed that similar extensions could form through AU or GU dinucleotides pairing with A1543 and U1544 of the aSD (Figure R2). This prediction has now been confirmed by our new *aur* cryo-EM structure.

We recognize that base-pairing flexibility likely exists at the 3' end of the 16S rRNA, encompassing the nucleotides that can form extended SD-aSD interactions. Although C1545 was observed flipped out in the *spa* structure (Figure R2), it may participate in

Watson-Crick pairing in other mRNAs. We therefore retained the predicted pairing through C1545 for *ccpN* mRNA.

For *cvfC1* mRNA, we proposed an extended SD-aSD alignment relying on its GU dinucleotide pairing with A1543 and U1544 rather than with C1545. This conformation was favored because it more closely matched our initial *spa* structural data (now confirmed by the new *aur* structure). In summary, all base-pairing schemes were proposed based on consistency with available structural data and biochemical validation, while acknowledging that multiple conformations are likely possible within the extended SD-aSD sequence space.

Figure R2. Extended SD–aSD interactions with structural support compared to predictions. Predicted and experimentally supported SD-aSD base-pairing schemes for representative mRNAs. The top panels show configurations supported by cryo-EM structures for *spa* and *aur* mRNAs, while the lower panels display predicted potential extensions for *ccpN* and *cvfC1*. Green, 16S rRNA aSD; blue, mRNA SD; dashed lines, proposed or observed base-pairs, start codons are underlined.

3. Based on the cryo-EM structure, the authors propose that the extended SD-helix results from two additional A-U basepairs with an unpaired C in the aSD and with a 4nt spacing from the SD to the UUG start codon. This may well be true, however, the way the complex has been formed raises many concerns about the relevance of this structure...One would have expected that the authors would have added the mRNA to the 70S ribosomes in the presence of fMet-tRNA, however, this appears not to be the case. Rather 70S ribosomes were incubated for 10mins at 37deg with 1mM magnesium (to split them?) and then mRNA and deacylated tRNA^{fMet} (why not fMet-tRNA?) and then non-aminoacylated tRNA^{Lys} – why was this added at all? And then subsequently the magnesium was increased to 15mM. It is unclear to me what to expect under these

conditions...do all 70S ribosomes get programmed with deacylated tRNA^{fMet} in the P-site and tRNA^{Lys} in the A-site, or can they also be programmed with tRNA^{Lys} in the P-site and tRNA^{fMet} in the E-site?

We thank the reviewer for raising these important points regarding the formation of the 70S initiation complex on *spa* mRNA. Below, we explain the rationale behind our experimental setup, clarify the purpose of each component, and describe how these concerns were addressed in our new *aur* mRNA complex.

The incubation of 70S ribosomes at 1 mM Mg²⁺ was intended to promote subunit “breathing” to facilitate mRNA loading. The subsequent increase of Mg²⁺ (to 15 mM) was to stabilize complex formation and reduce sample heterogeneity. Based on extensive optimization in our laboratory, this protocol reproducibly yields well-formed *S. aureus* 70S complexes suitable for cryo-EM, and is more efficient than assembling 30S and 50S subunits separately.

Because initiation factors were omitted, deacylated tRNA^{fMet} was used as an efficient substrate to occupy the P-site and stabilize initiation-like 70S complexes. This was established in previous structural studies, indicating no difference between acylated and deacylated tRNA^{fMet} in the positions of their anticodon in the decoding site of the 30S subunit (e.g., Agrawal et al., 1999; Yusupova et al., 2001 and Jenner et al., 2005). In the original *spa* complex, non-aminoacylated tRNA^{Lys} was additionally supplied to occupy the A-site, a standard practice to enhance ribosomal stability and reduce conformational flexibility. To directly address the reviewer’s concern, we omitted all tRNAs other than deacylated tRNA^{fMet} in the preparation of the new *aur* 70S complex.

The resulting *aur* mRNA complex fully confirms the extended SD-aSD interaction and strengthens the mechanistic conclusions of our study.

References:

Agrawal RK, Penczek P, Grassucci RA, Burkhardt N, Nierhaus KH, Frank J. Effect of Buffer Conditions on the Position of tRNA on the 70S Ribosome as Visualized by Cryoelectron Microscopy. *Journal of Biological Chemistry*. 1999;274(13):8723-9.

Yusupova GZ, Yusupov MM, Cate JH, Noller HF. The path of messenger RNA through the ribosome. *Cell*. 2001;106(2):233-41.

Jenner L, Romby P, Rees B, Schulze-Briese C, Springer M, Ehresmann C, et al. Translational operator of mRNA on the ribosome: how repressor proteins exclude ribosome binding. *Science*. 2005;308(5718):120-3.

The methods on page 25 suggest that the major states are “ (1) non-rotated with E-site tRNA, (2) rotated 70S with A-site tRNA, (3) rotated 70S with A- and P-site tRNAs...” however, the actual number of particles needs to be detailed here. Also, it is hard to envisage rotated ribosomes with canonical tRNAs...presumably they are hybrid given that deacylated tRNAs were used? Also, it is hard to imagine a state with only A-tRNA and no P-tRNA? In this regard, the text appears to differ from the Sup Fig 6 which suggests that the classes that were merged contained P- and A-tRNAs, and P- and E-tRNAs although here it is also not mentioned whether they are rotated or not?

We thank the reviewer for carefully examining the technical aspects of our cryo-EM data and for spotting some inaccuracies in the Materials and Methods section. Indeed, there was an error in the description of the image processing data for the *spa* mRNA 70S complex. The correct major classes obtained were:

- (1) rotated 70S with E-site tRNA,
- (2) non-rotated 70S with E- and P-site tRNAs,
- (3) non-rotated 70S with E-, P-, and A-site tRNAs, and
- (4) 50S subunits.

These states are consistent with the data presented in the original Supplementary Figure S6. We apologize for the confusion caused by the incorrect description in the methods.

In the revised version of the manuscript, we have now replaced the previous *spa* complex with the newly obtained 70S initiation complex programmed with *aur* mRNA. The new structure directly complements our biochemical and *in vivo* reporter analyses, providing a coherent mechanistic framework that integrates structural, biochemical, and functional data. It also resolves several of the specific concerns raised by Reviewer 1 regarding the composition and assembly of the original 70S initiation complex (as detailed above). In this new analysis, we accurately describe all particle classes, their respective tRNA occupancies, and rotational states. Briefly, tRNA^{fMet} is in P-site, the 70S is non-rotated, and, as expected, no class with isolated A-site tRNA was detected. All relevant details, including particle numbers per class and image-processing parameters, have been added to the revised *Methods* section.

This new structural dataset fully resolves the ambiguities identified by the reviewer and provides a more precise basis for interpreting extended SD-aSD interactions in *S. aureus*.

Since the accompanying map appears to have density for A-, P- and E-tRNAs, one can presume that these later (non-rotated) states were merged. This raises the question whether the authors could be sure that the P-tRNA in both cases is actually the initiator

tRNA? This is important since if the P-tRNA in one case was the tRNA^{Lys} with the initiator tRNA in the E-site, then the authors have merged two states with different linker lengths to the extended Shine-Dalgarno helix. It would seem appropriate to convince the reader that these substates are the same before merging.

In the originally submitted *spa* complex, deacylated tRNA^{fMet} occupied the P-site and non-aminoacylated tRNA^{Lys} was added to stabilize the A-site. We acknowledge that the presence of tRNA^{Lys} could have theoretically introduced heterogeneity during data processing. To eliminate this possibility entirely, no additional tRNA species besides initiator tRNA were used in forming the new *aur* mRNA complex.

The cryo-EM map provided has been highly filtered and very tightly masked – it would seem appropriate for the authors to provide the unsharpened 3d refine and post-processed maps so one can assess the correlation between the map and models before the sharpening process has occurred.

In the previous submission, the Figshare repository contained only the filtered map obtained through the EMReady application from CryoSPARC. For the newly obtained *aur* mRNA structural data, we have now uploaded the CryoSPARC refined map without use of EMReady. This is the map used for model building and validation, and it has also been deposited in the PDB together with the refined coordinates.

4. Overall, it is unclear why the authors also only collected 1105 movies, yielding only 150K particles and limiting themselves to lower resolution – especially given that it is important for the study to understand exactly which tRNAs are bound in which positions so as to have the register and resolution to know how the SD and aSD are basepaired. I am not convinced this is possible with the resolution that they report.

We agree with the reviewer that the number of movies collected for the initially submitted *spa* mRNA 70S complex restricted the achievable resolution. While the previous dataset was sufficient to demonstrate the principle of extended SD-aSD formation, we recognize that higher resolution is important to unambiguously define tRNA positions and base-pairing interactions.

To address this and the other points raised by reviewer 1, we prepared a new initiation complex programmed with *aur* mRNA and performed an expanded cryo-EM data collection. For this dataset, 11,450 movies were recorded, yielding a substantially larger pool of particles. Image processing identified major classes corresponding to (1) rotated 70S with E-site tRNA (516,117 particles), (2) non-rotated 70S with tRNAs in E- and P-site (383,073 particles), (3) non-rotated 70S with tRNAs in E- and partially occupied P-site

(242,930 particles), (4) 50S (170,566 particles) and (5) “junk” (60,870 unaligned particles). Particles from classes 2 and 3 were further refined through focused 3D classification using a mask around the P-site tRNA and the SD-aSD helix, resulting in a final set of 316,031 particles. Non-uniform refinement with activated defocus and global CTF refinement produced a 2.3 Å map, as determined by Fourier shell correlation.

This higher-resolution structure now provides extensive details for the interpretation of extended SD-aSD interactions and tRNA positioning within the *S. aureus* 70S initiation complex.

5. The additional base-pairing formed in *Staphylococcus* is proposed to be possible because of the lack of S21, and that the presence of S21 in *E. coli* prevents this extended helix forming. This is not taken to its conclusion by demonstrating that *E. coli* 70S lacking S21 can also form extended helices and therefore initiate translation as observed in *Staphylococcus*.

We thank the reviewer for this insightful comment. As correctly noted, we proposed in the Discussion that the absence of stably associated S21 in *S. aureus* ribosomes could allow formation of the extended SD-aSD helices, whereas its presence in *E. coli* may sterically hinder these interactions.

We agree that directly testing this hypothesis, by generating *E. coli rpsU* mutants, purifying S21-deficient ribosomes, and analyzing initiation complex formation with or without reconstituted S21, would provide more definitive evidence for its role. However, these experiments represent a substantial and technically demanding project of their own, requiring extensive biochemical and structural work. As our present study focuses on defining the mechanistic basis of translation initiation site recognition in *S. aureus*, we consider this investigation beyond its scope, though it remains a central direction for future research in our laboratory.

6. While the authors convincingly show that *rbfL* is involved in regulating the expression of the downstream *rbf* gene, some questions still remain. The authors propose that the presence of rare AGA and AGG codons in the *rbfL* uORF causes stalling under poor nutrient conditions. This is support by experiments in Figure 6C-G.

It is unclear to me why the authors didn't simply substitute the rare arginine codons for the others ones (CGU, CGC, CGA, CGG) but rather mutate one of them to a glycine codon?

We thank the reviewer for this insightful question. As shown in Figure R3 below, the coding sequence of the *rbfL* uORF is tightly intertwined with the RBS of *rbf*, making it

highly sensitive even to single nucleotide substitutions. Both rare arginine codons (AGA and AGG) contribute to the predicted SD motif of the *rbf* RBS, providing base pairing interactions with the 16S rRNA 3' end.

We initially focused on the second arginine codon (AGG) because our ribosome profiling data indicated that AGG codons are decoded more slowly than AGA codons, based on pause scores from several ribosome profiling experiments performed in our laboratory. Since the second rare Arg codon (AGG) also forms the core of the *rbf* SD motif, it was a particularly interesting target for testing the coupling between *rbfL* translation and *rbf* translation initiation. Substitution with another arginine codon (e.g., CGG) could, in theory, preserve the SD-like sequence but would introduce an even rarer codon decoded by the tRNA^{Arg(CCG)}, the least abundant tRNA in *S. aureus* according to our nanopore-based tRNA quantification (Pre-print: Jaramillo-Ponce., 2025). Therefore, one of few viable mutations that preserve a strong SD motif was the replacement of AGG with a glycine GGG codon, which still allows potential G:U wobble pairing with the aSD sequence. Furthermore, we have now generated additional mutants replacing the AGG with a glutamate (GAG) codon, maintaining an SD-motif (GAGG > GGAG). Both mutations abolished *rbfL*-dependent repression and rendered *rbf* insensitive to L-Arg supplementation (Supplementary figure S6D).

Figure R3: Contribution of the rare arginine codons in *rbfL* to the Shine–Dalgarno motif of the downstream *rbf* TIS. The two rare arginine codons (AGA and AGG) within the *rbfL* uORF overlap with the predicted SD sequence of the downstream *rbf* gene. The second rare codon (AGG) forms the core of the SD-like motif, while the first (AGA) also contributes to potential base-pairing interactions with the 16S rRNA aSD. Because of this overlap, most single-nucleotide substitutions simultaneously alter both the *rbfL* coding sequence and the *rbf* ribosome-binding site, making functional interrogation of individual codons highly constrained.

Because these changes increased *rbf* translation irrespective of uORF translation or arginine levels, they uncoupled *rbf* from *rbfL*, consistent with *rbfL*-dependent control, but not alone proving the arginine codon's necessity. Following the reviewer's suggestion, we therefore mutated the first rare arginine codon (AGA) to a more commonly used, non-rare and efficiently decoded arginine codon (CGU). While this change abolished the arginine

dependence of *rbf* translation, it also resulted in a ~300-fold decrease in its overall expression, likely reflecting a disruption of the finely balanced *rbf* TIS structure and a weakened SD motif.

Thus, the *rbf* TIS sequence appears to be finely tuned to allow for similar translation initiation efficiency of both *rbfL* and *rbf* (Figure 6B), facilitating *rbfL* translation dependent regulation of *rbf*. This balance is very sensitive to nucleotide changes, making it difficult to assess individual amino acid or codon contributions.

Especially due to the difficulty with *rbfL* sequence changes, we had tested whether limited charging of tRNA^{Arg(UCU)} contributed to the regulatory mechanism by overexpressing this tRNA, which consistently yielded a partial derepression of *rbf* translation when cells are grown in RPMI (Figure 6F).

Reference:

Jaramillo-Ponce JR, Wolff P, Marchand V, Motorin Y, Kohl M, Kanazawa H, et al. Complete post-transcriptional modification profiles in individual *Staphylococcus aureus* tRNA species. *bioRxiv*. 2025:2025.10.30.685614.

Its also unclear to me why addition of extra L-Arg should help translation through rare codons - or are the authors proposing that there is also some charging problem due to Arg limitation?

We thank the reviewer for raising this point. Yes, our model proposes that arginine limitation reduces tRNA^{Arg(UCU)} charging, thereby slowing decoding of the two rare arginine codons within *rbfL*. We had stated this in the original Results section (“*We hypothesized that limited availability of the charged tRNA^{Arg(UCU)} could constrain rbfL translation, thereby modulating downstream rbf expression.*”) and in the Discussion, where we described ribosome pausing on the rare codons when arginine is limiting (“*Strikingly, we show that a novel leader peptide, rbfL, senses arginine limitation through the presence of a rare arginine codon, which is decoded by the low-abundance tRNA^{Arg(UCU)}. When arginine is scarce, ribosomes pause or stall on rbfL, blocking the ribosome binding site of the downstream gene rbf, and preventing its translation initiation.*”).

To further clarify, we have now revised the Discussion to explicitly highlight the role of tRNA charging levels: “*We show that a novel leader peptide, rbfL, senses arginine limitation through the presence of rare arginine codons, which are decoded by the low-abundance tRNA^{Arg(UCU)}. When arginine is scarce and tRNA charging levels decrease,*

ribosomes pause on rbfL and block the ribosome binding site of the downstream gene rbf, preventing its efficient translation initiation.”

This is supported by our reporter assays supplementing L-arginine or over-expressing the tRNA^{Arg(UCU)} in the poor RPMI medium, relieving repression of *rbf*. Our rationale for this interpretation is that tRNA charging levels, including tRNA^{Arg(UCU)}, decrease under nutrient limitation. Subsequently, the two arginine codons within *rbfL* are slowly decoded and pause ribosomes co-translationally. Our own nanopore-based tRNA quantification of *S. aureus* (Pre-print: Jaramillo-Ponce et al. 2025) revealed that tRNA^{Arg(UCU)} is a low abundant tRNA isoacceptor species. Under nutrient limiting conditions, it competes with the other tRNA isoacceptors for charging. On one hand, replenishing the pool of available L-Arginine is expected to fully restore charging levels. On the other hand, increasing the levels of the tRNA^{Arg(UCU)}, is expected to improve its ability to compete with the more abundant tRNA isoacceptors and to increase the amount of charged tRNA^{Arg(UCU)} in the cell. Indeed, such effects are well documented, including for rare arginine codons, where it could be shown that *argU* (encoding tRNA^{Arg(UCU)}) co-expression can improve recombinant protein expression in *E. coli* (i.e, Garcia et al., 1996).

References:

Garcia OL, González B, Menéndez A, Sosa AE, Fernández JR, Santana H, et al. The *argU* gene product enhances expression of the recombinant human alpha 2-interferon in *Escherichia coli*. *Ann N Y Acad Sci*. 1996;782:79-86.

Jaramillo-Ponce JR, Wolff P, Marchand V, Motorin Y, Kohl M, Kanazawa H, et al. Complete post-transcriptional modification profiles in individual *Staphylococcus aureus* tRNA species. *bioRxiv*. 2025:2025.10.30.685614.

The Y-axis of graphs C-G are all different making it hard to compare between conditions – maybe one should not but still it would be better to have the same axis values.

Because these experiments were performed in different media (e.g., BHI and RPMI), the luminescence values of the Firefly/Renilla reporter fusions differ substantially across conditions. Using a uniform Y-axis would therefore compress some datasets and obscure relevant differences within each experiment. For this reason, we kept condition-specific Y-axis scales.

To facilitate comparison across conditions, we have now added an additional panel (new Figure 6G) that presents fold-changes in *rbf* expression with or without *rbfL* translation. This standardized comparison allows the reader to directly assess the regulatory effect of the uORF under each growth condition, independently of baseline reporter activity.

As part of this reorganization, the data from the arginine to glycine codon substitution has been moved to the supplementary material (Figure S6D), alongside the new data for the arginine to glutamate substitution.

The rescue by 10X L-Arg is better than by expression of the tRNA^{Arg(UCU)} – this doesn't appear to really support the rare Arg codon theory?

On this particular point, we respectfully disagree with the reviewer's interpretation and believe that the observed difference between L-Arg supplementation and tRNA^{Arg(UCU)} overexpression actually supports the rare-codon mechanism.

Supplementing 10× L-Arg directly increases the intracellular amino acid pool and therefore fully restores charging of all arginine tRNA isoacceptors, including the low-abundance tRNA^{Arg(UCU)}. Under these conditions, *rbfL* is translated efficiently, and repression of *rbf* is relieved, exactly as predicted for a regulatory mechanism that depends on reduced charging of a scarce tRNA (Figure 6E).

In contrast, overexpressing tRNA^{Arg(UCU)} increases the amount of this isoacceptor but does not solve the underlying limitation in arginine availability. Therefore, a higher concentration of tRNA^{Arg(UCU)} can improve the competition against the other isoacceptors and increase the total amount of charged tRNA^{Arg(UCU)} in the cell. However, this is still limited by the restricted amount of L-Arg available for charging. Therefore, *argU* overexpression is expected to only partially relieve *rbfL*-dependent repression, which we observe experimentally (Figures 6F, 6G). In conclusion, these results are consistent with a model in which both amino acid availability and isoacceptor abundance shape *rbfL* translation efficiency (and its impact on *rbf* translation).

7. Finally, the overall model for *rbfL* regulation of *rbf* expression is that stalling near the stop codon of *rbfL* prevents access of 30S subunits to the SD and start codon of the downstream *rbf* gene, however, the authors do not demonstrate that ribosomes stall, let alone where they stall in the *rbfL* ORF. This would seem like an important experiment to perform before one can make such a claim as shown in Figure 6H.

We thank the reviewer for raising this important point. In the original manuscript we had intentionally used the wording “*stall or pause*” because we had not directly distinguished between these two possibilities experimentally. To clarify the mechanism, we have now performed additional co-translational toeprinting assays to determine whether ribosomes stall or pause during translation of the *rbfL* leader peptide (new Supplementary Fig. S6A-B). Using the Δ ribosome PURE system, supplemented with *S. aureus* 70S, we monitored ribosome movement on the wild-type *rbfL* sequence and compared it to two controls:(i)

Retapamulin (Ret), which traps ribosomes at the start codon, and (ii) Apidaecin-137 (Api), which traps ribosomes at the stop codon. As shown in Supplementary Fig. S6A, translation of the wild-type *rbfL* does not produce a strong single toeprint that would indicate a defined stall site. Instead, several positions along the ORF produce toeprint signals consistent with co-translational pausing. These co-translational pauses are dependent on elongating 70S, because they are lost upon treatment with Ret or Api. Notably, two prominent pauses coincide with the position +13 relative to the rare Arg codons AGA and AGG at the expected distance for decoding in the A-site. Notably, these pauses arise during *in vitro* translation under optimal conditions, with abundant amino acids and total *E. coli* tRNAs. Importantly, tRNA^{Arg(UCU)} is also among the least abundant Arg tRNAs in *E. coli*. Therefore, these slow-decoding events are fully consistent with the behavior expected for rare codons and would be further exacerbated *in vivo* under arginine-limiting conditions when tRNA^{Arg(UCU)} charging levels decrease.

To confirm that the pauses depend on the rare Arg codons, we conducted an alanine-scanning mutational analysis substituting the amino acids of the RKR stretch, encompassing the two Arg codons (Supplementary Fig. S6B). Specifically, when mutating the Arg codons, the corresponding toeprint signals disappeared, demonstrating that both rare codons are required for the observed pausing signature.

These new results support several key conclusions:

1. *rbfL* does not induce ribosome stalling per se. No stalling was detected *in vitro* unless a stalling-inducing ligand (e.g., Retapamulin) was added.
2. Translation of *rbfL* inherently contains slow-decoding steps at the two rare arginine codons.
3. Under arginine-limiting conditions, reduced tRNA^{Arg(UCU)} charging levels are expected to increase the duration of pausing and thereby enhance occlusion of the downstream *rbf* ribosome-binding site.
4. Pausing, rather than stalling, fully explains the regulatory behavior captured in our *in vivo* reporter assays.

While we cannot exclude that additional factors might trigger bona fide stalling *in vivo*, the combined biochemical, genetic, and reporter data strongly support a model in which pausing at rare Arg codons provides the regulatory signal controlling access to the *rbf* start site.

Some additional minor points:

1. Abstract line 26: What is striking about it compared to the many known examples? This is quite subjective, perhaps remove the word striking?

The word 'striking' has now been replaced by the word 'novel'.

2. Line 51: Perhaps also include citation to PMID: 17355865 from Yokoyama group that showed SD interaction on 30S subunit, rather than 70S, at higher resolution than reference 13 also in the same year (2007)

We thank the reviewer for the suggestion to include this relevant citation, which has now been added.

3. Line 75: I thought reference 46 showed that they could obtain high resolution despite using MNase?

We thank the reviewer for this comment. In the original ref 46, Allen Buskirk and coworkers demonstrated that relatively high resolution can be achieved in *E. coli* despite using MNase, thanks to substantial optimization of cell harvesting and lysis conditions. However, as the authors note themselves, MNase has an inherent sequence bias that fundamentally limits the attainable resolution in bacterial Ribo-seq, even under optimal conditions. In *E. coli*, RNase 1 cannot be used because it is rapidly inhibited by components of the cell extract or by *E. coli* ribosomes. Therefore, they maximized the resolution attainable with MNase, which however remains lower than what could be achieved with RNase 1 in organisms where it is compatible.

In our work with *S. aureus*, RNase 1 is fully compatible with cell extracts, allowing us to apply the similarly optimized lysis principles (e.g., high-Mg²⁺ polysome stabilization) while benefiting from RNase 1's unbiased cleavage pattern. This produces substantially higher resolution in both Ribo-seq and Ribo-RET than with MNase-based approaches. Indeed, the limitations of MNase bias have motivated efforts to computationally correct it (Zhao et al., 2019), underscoring why RNase 1 provides a technical advantage wherever it can be used.

Reference:

Zhao D, Baez WD, Fredrick K, Bundschuh R. RiboProP: a probabilistic ribosome positioning algorithm for ribosome profiling. *Bioinformatics*. 2019;35(9):1486-93

Reviewer #2 (Remarks to the Author):

In bacteria, a major cis-acting sequence element influencing translation initiation is the Shine-Dalgarno (SD) sequence, which facilitates ribosome binding upstream of start codons. Building on this concept, this manuscript presents a high-resolution analysis of translation initiation in *Staphylococcus aureus*, a clinically relevant pathogen. Using RNase I-based ribosome profiling in combination with Retapamulin treatment (Ribo-Ret), the authors accurately map translation start sites across the *S. aureus* transcriptome and also monitor the usage of rare, unconventional start codons. In addition, the authors identify numerous previously unannotated small open reading frames (ORFs), including regulatory upstream ORFs (uORFs).

A key finding is that *S. aureus* initiation involves extended Shine-Dalgarno motifs, which form strong base-pairing interactions with the anti-SD region of the 16S rRNA. Notably, the study identifies two non-canonical start codons, AUU and AUA, whose recognition depends on these extended SD sequences. The authors further investigate a biologically significant case of uORF-mediated regulation: translation of a small leader peptide modulates expression of *rbf*, a transcription factor essential for biofilm formation. This regulatory mechanism is conditionally governed by codon usage bias and arginine availability.

Overall, this study reveals key differences from canonical translation initiation mechanisms described in *E. coli*, highlighting unique aspects of translational regulation in *S. aureus*. In this regard, the work constitutes a significant contribution to the field and deserves recognition, even if certain ambiguities and limitations remain.

We thank reviewer 2 for recognizing the significance of our work and for providing valuable feedback to improve our manuscript.

Major comments:

1- Line 52–53: The sentence discussing the S1 ribosomal protein is very vague. **The specific role or relevance of this protein in the context of the study should be clarified.**

We thank the reviewer for pointing out the need for clarification regarding ribosomal protein bS1. In the Introduction, we briefly describe the role of bS1 in *E. coli*, where it promotes initiation by binding single-stranded mRNA regions and helping to unwind secondary structures near the start codon. In the original Discussion, we contrast this with

S. aureus, where bS1 is non-essential and not stably associated with the ribosome (Khusainov et al., 2016), suggesting that its contribution to initiation is likely very different.

Our laboratory is actively investigating the precise functions of bS1 in *S. aureus*, which remain largely unknown. Given that a full characterization of bS1 in *S. aureus* is beyond the scope of the present manuscript, we now clarify its role without overstating conclusions and highlight it as an open question for future work. Therefore, we have changed the Discussion accordingly:

“Another important difference is the absence of a ribosome-anchored bS1 protein in *S. aureus* (Khusainov et al., 2016). While bS1 is essential in *E. coli* and facilitates initiation by promoting mRNA loading and resolving structured ribosome-binding sites (Duval et al., 2013 and Webster et al., 2024), its function in *S. aureus* is poorly understood and appears to diverge substantially. Further work will be required to clarify the respective contributions of bS21 and bS1 to translation initiation in *S. aureus*.”

References:

Khusainov I, Vicens Q, Bochler A, Grosse F, Myasnikov A, Ménétret J-F, et al. Structure of the 70S ribosome from human pathogen *Staphylococcus aureus*. *Nucleic Acids Research*. 2016;44(21):10491-504.

Duval M, Korepanov A, Fuchsbaauer O, Fechter P, Haller A, Fabbretti A, et al. *Escherichia coli* ribosomal protein S1 unfolds structured mRNAs onto the ribosome for active translation initiation. *PLoS Biol*. 2013;11(12):e1001731.

Webster MW, Chauvier A, Rahil H, Graziadei A, Charles K, Miropolskaya N, et al. Molecular basis of mRNA delivery to the bacterial ribosome. *Science*. 2024;386(6725):eado8476.

2- Line 84–86 as well as throughout the manuscript: This analysis raises the question of how different the ribosomes of *E. coli* and *S. aureus* are, also with regard to the structure obtained through cryo-EM (Figure 3B) It would be helpful to briefly describe them or clarify whether such differences were systematically analyzed.

A systematic comparison of *E. coli* and *S. aureus* ribosomes has previously been published (Khusainov et al., 2016). In our Discussion we have already summarized the specific architectural differences most relevant to translation initiation and to our structure:

- The 3' end of 16S rRNA (aSD) is conserved, so other features must account for the distinct initiation rules.
- bS21: In *E. coli*, bS21 is longer and stably anchored near the SD-aSD helix; placing the *E. coli* bS21 into our *S. aureus* 70S initiation map produces a steric clash with

the shifted, extended SD-aSD duplex. In Firmicutes, bS21 is shorter and lacks C-terminal residues required for tight anchoring (Jha et al., 2020), consistent with no density for bS21 in our map.

- bS1: Unlike *E. coli*, *S. aureus* lacks a stably ribosome-anchored bS1, further suggesting species-specific differences in mRNA recruitment.
- Our goal here is not a full systematic inventory, but a targeted structural rationale for extended SD-aSD accommodation in *S. aureus*.

References:

Khusainov I, Vicens Q, Bochler A, Grosse F, Myasnikov A, Ménétret J-F, et al. Structure of the 70S ribosome from human pathogen *Staphylococcus aureus*. *Nucleic Acids Research*. 2016;44(21):10491-504.

Duval M, Korepanov A, Fuchsbaauer O, Fechter P, Haller A, Fabbretti A, et al. *Escherichia coli* ribosomal protein S1 unfolds structured mRNAs onto the ribosome for active translation initiation. *PLoS Biol*. 2013;11(12):e1001731.

Webster MW, Chauvier A, Rahil H, Graziadei A, Charles K, Miropolskaya N, et al. Molecular basis of mRNA delivery to the bacterial ribosome. *Science*. 2024;386(6725):eado8476.

Why does an extended basepairing between SD and anti-SD necessitate a longer linker sequence upstream of the start codon in *S. aureus* (2A)?

We appreciate the request for clarification. The extended SD-aSD helix we observe in *S. aureus* shifts the anchoring point of the mRNA on the 30S platform. In our cryo-EM map, the SD-aSD duplex sits further along the mRNA path toward uS2 and away from uS11, where it is stabilized by contacts from uS2 and bS18. Because the helix is shifted and includes extra base pairs immediately proximal to the start codon, nucleotides that would otherwise belong to the spacer are effectively used for pairing. To keep the start codon correctly positioned in the P site, the optimal SD-start spacing becomes longer when measured from the canonical SD core. In other words, the increased spacing is not a passive “linker extension,” but the structural consequence of the extended SD-aSD duplex that engages bases nearer the start codon. In the results section of our manuscript, we have explained this with the following updated paragraph:

‘This extended duplex occupies a large region of the ribosomal platform, spanning further along the mRNA path, from uS11 toward uS2 (Figure 3A, C). Positively charged residues of uS2 and bS18 contact the backbone of the 16S rRNA 3’ end and stabilize this shifted position of the SD-aSD helix (Figure 3C). The additional base-paired AU dinucleotide corresponds to the increased spacing between SD and start codon in S. aureus that we

found (Figure 2A), and appears to strengthen the SD-aSD interactions, i.e., the sequence insertion is not a linker increase but rather it provides additional complementary base pairing opportunities.'

3- Lines 114–121: It appears that the authors refer to the “no drug” condition as a Ribo-seq dataset. However, capturing ribosome-protected fragments typically involves an elongation-stalling antibiotic to arrest translating ribosomes. Could the authors clarify how ribosome footprints were obtained in the absence of such treatment?

We appreciate the opportunity to clarify terminology. In our manuscript, “Ribo-seq” refers to the antibiotic-free condition, used in contrast to Ribo-RET (Retapamulin-treated). Following recommendations from published Ribo-seq methods by the laboratories of Allen Buskirk, Cynthia Sharma and Alexander Mankin, we developed a protocol suited for *S. aureus* Ribo-seq (Kohl et al., 2024 and Figure R4 below). Indeed, we did not use an elongation inhibitor for standard Ribo-seq, but we stabilized translating ribosomes by optimizing harvest and lysis conditions, as follows:

- Rapid cooling and harvest: Cultures were swirled within an ice bath and immediately harvested by centrifugation to minimize elongation after sampling.
- High-Mg²⁺ lysis in presence of non-hydrolyzable GTP analog: cells were bead-lysed in buffer containing 50 mM MgCl₂ and 1 mM GMPPNP, which jointly suppress subunit dynamics and translocation in the lysate, effectively arresting elongation without antibiotics.
- Single-buffer RNase 1 digestion: RNase I digestion was performed in the same high-Mg²⁺/GMPPNP buffer, avoiding a sucrose-cushion step prior to nuclease treatment. This was important because pelleting can strip A-/P-site tRNAs and open the mRNA channel, compromising read-length/position resolution.

These conditions yielded high-quality footprints with sharp read-length distributions and expected metagene features (Figure 1), validating that the “no-drug” datasets represent bona fide Ribo-seq profiles. To avoid confusion, we have revised figure legends to state explicitly whether datasets are Ribo-seq (no drug) or Ribo-RET (Retapamulin).

Figure R4: Workflow for antibiotic-free Ribo-seq and Ribo-RET in *S. aureus*. Cells are rapidly chilled and harvested, bead-lysed in 50 mM MgCl₂ + 1 mM GMPPNP to arrest elongation post-lysis, followed by RNase 1 digestion in the same buffer. Ribo-RET samples additionally receive Retapamulin prior to harvest to enrich initiation complexes. Footprints are purified by size selection and used for library construction.

Reference:

Kohl MP, Chane-Woon-Ming B, Bahena-Ceron R, Jaramillo-Ponce J, Antoine L, Herrgott L, et al. Ribosome Profiling Methods Adapted to the Study of RNA-Dependent Translation Regulation in *Staphylococcus aureus*. In: Arluison V, Valverde C, editors. *Bacterial Regulatory RNA: Methods and Protocols*. New York, NY: Springer US; 2024. p. 73-100.

4- Line 116: The observation of larger ribosome footprints is indeed intriguing. While the authors propose that this may be due to extended SD–anti-SD interactions, it is noted that only 28% of mRNAs possess such extended SD motifs (Figure 3F). This proportion seems insufficient to account for the dominant larger peak observed in Figure 1D. Is this distribution also seen in comparable experiments in other species, and which other factors may contribute to this footprint size distribution?

We appreciate the opportunity to clarify. The large footprints we report under Ribo-RET (shift from ~26-29 nt to ~35-38 nt; Fig. 1D) reflect initiation-specific protection that arises from SD-aSD pairing in general, not solely from *extended* SD motifs. This interpretation is consistent with prior work showing that SD-aSD base pairing increases protection from nuclease cleavage during initiation and can enrich long reads at SD sites (e.g., Buskirk lab studies).

A bimodal footprint-size distribution is commonly observed in bacteria, but the peak positions depend on protocol (nuclease, Mg²⁺, buffer, timing). For example, *Salmonella* and *Listeria* Ribo-RET datasets (ref 47) report initiation peaks around ~28-31 nt with shorter elongation peaks (~24 nt), illustrating species- and protocol-dependent differences.

Regarding the 28-30% estimate for *extended* SD sites:

- This is a conservative lower bound based on interactions we could validate structurally/biochemically (e.g., *aur*, *spa*).
- The global Ribo-RET length shift does not require that most mRNAs carry extended SDs; canonical SD-aSD pairing alone increases protection.
- That said, the magnitude of the length increase in *S. aureus* likely reflects a combination of factors: (i) a shifted SD-aSD anchoring point on the platform seen by cryo-EM, (ii) additional start-proximal base pairs (extended interactions) in a subset of mRNAs, and (iii) the longer average SD-start spacing we measure in *S. aureus*.

We have revised the Results to clarify that (i) long footprints in Ribo-RET primarily report SD-aSD protection during initiation (extended or canonical), and (ii) our ~30% figure reflects validated extended cases, with the true fraction likely higher given plausible alternative extended alignments not yet resolved structurally. The text has been modified as follows:

‘The stark increase in read length upon Ret treatment can be rationalized by the additional protection that SD-aSD base pairings confer during bacterial initiation (Mohammad et al., 2016).’

'While we predict the presence of canonical SD-sequences in approximately 84% of S. aureus mRNAs based on computational SD-motif search, our analysis indicates a conservative estimate of approximately 30% of initiation sites relying on extended SD-aSD base pairing (Figure 3G). This prediction is based on the experimentally validated base pairing possibilities of the aur mRNA and does not include other alternative interactions, which we consider a likely possibility.'

Reference:

Mohammad F, Woolstenhulme CJ, Green R, Buskirk AR. Clarifying the Translational Pausing Landscape in Bacteria by Ribosome Profiling. Cell Reports. 2016;14(4):686-94.

5- Figure 1E (and similar graphs): It would be more intuitive to present the sequence in the graph in the 5'→3' orientation to facilitate sequence comparisons.

We appreciate the suggestion. To improve readability, we have added explicit 5'→3' direction arrows next to all sequence tracks and Ribo-seq/Ribo-RET profiles in Figure 1E and throughout the manuscript. We retain the plotting in genomic strand orientation so that footprint peaks align with their native coordinate system (useful for cross-referencing annotations and operon context).

6- Figure 1E: It is unclear why the upper panel displays the RNase I control and RNase I + Retapamulin datasets, while the zoom-in panel below shows the MNase + Retapamulin and RNase I control datasets. For clarity and consistency, it could be an option to show all datasets?

Thank you for the suggestion. We have revised Figure 1E to include all four conditions side-by-side: MNase (no drug), MNase + Ret, RNase 1 (no drug), RNase 1 + Ret. The zoomed-in panel now focuses on the initiation region and displays both MNase + Ret and RNase 1 + Ret for a direct comparison of initiation peak shape and positional precision.

All labels have been updated to clearly indicate Ribo-seq (no drug) vs Ribo-RET (Retapamulin), the nuclease used, and the RPF size windows are shown.

Is the size bar in the upper panel (ctrl) correctly labelled, as it differs significantly from the others?

Yes, the scale bars are correct. The larger amplitude in the Retapamulin (Ribo-Ret) traces reflects the expected enrichment of ribosomes at start codons when initiation is arrested,

leading to a several-fold increase in footprint density relative to the no-drug control (nearly ten-fold in this example for *sORF20*).

7- Figure 2D: The luciferase assay suggests that *S. aureus* 70S ribosomes prefer both extended SD sequences and GUG as a start codon. It would be interesting to test whether moving the GUG start codon closer to the SD (e.g., in a GUGAUG context) changes the result in the toeprinting assay. Some studies (e.g., Yamamoto et al., 2016, PNAS) suggest ribosomes may favor the first start codon from the 5' end. Is initiation mainly driven by the SD interaction, or also by start codon identity and in-frame position?

Thank you for the clarification request. For canonical SD-led *de novo* initiation, our data indicate that SD-start aligned spacing is the determinant of start-site choice. In our previous study (Kohl et al., 2022) and in the present work, we did not detect an intrinsic bias for AUG vs GUG once spacing places the triplet optimally in the P site. In *S. aureus*, the optimal spacing window is right-shifted relative to *E. coli*, consistent with the shifted/extended SD-aSD duplex observed structurally. In the *aur* spacer series (Fig. 3D-E), changing spacing by two nucleotides switches decoding between the overlapping codons, illustrating spacing-driven selection.

Regarding Yamamoto et al. (2016), the “70S-scanning” mechanism pertains to post-termination scanning in polycistronic contexts and is distinct from the mono-cistronic, SD-led, *de novo* initiation sites with potential start site ambiguity we analyzed. Thus, the “first start from the 5' end” rule does not apply here.

We did not introduce an additional GUGAUG swap in this revision, but we believe that reversing the order would yield the predicted spacing-dependent outcome, consistent with Fig. 3D-E.

References:

Kohl MP, Kompatscher M, Clementi N, Holl L, Erlacher Matthias D. Initiation at AUGUG and GUGUG sequences can lead to translation of overlapping reading frames in *E. coli*. *Nucleic Acids Research*. 2022;51(1):271-89.

Yamamoto H, Wittek D, Gupta R, Qin B, Ueda T, Krause R, et al. 70S-scanning initiation is a novel and frequent initiation mode of ribosomal translation in bacteria. *Proceedings of the National Academy of Sciences*. 2016;113(9):E1180-E9.

8- Figure 3D: A brief interpretation of the *E. coli* results with regard to the impact of the SD sequence would be helpful.

We thank the reviewer for highlighting the need for additional interpretation of the *E. coli* data in original Figure 3D (now Figure 3E). We have revised the text to improve clarity, now stating:

'In E. coli, all variants were decoded at the first initiation triplet (AUG) irrespective of spacer mutations, consistent with the absence of extended SD-aSD accommodation. The tested spacers placed AUG more optimally in the P-site of E. coli ribosome, rendering start-site choice largely insensitive to local RBS changes that shift decoding in S. aureus.'

9- Figure 3F–G: The figures show the percentage of mRNAs with extended SD sequences and start codon usage. Could the authors also indicate which start codons most often co-occur with extended SD motifs?

We have expanded the analysis to relate start-codon identity to (i) presence of any SD motif and (ii) presence of extended SD motifs (Figure R5 below).

All annotated ORFs (genome-wide):

- Any SD motif % by start codon: AUG 87.7%, GUG 78%, UUG 63%. The lower SD frequency for UUG likely reflects annotation bias and/or non-expression under our conditions.
- Extended SD usage % by start codon: AUG 29.4%, GUG 27.1%, UUG 20.8%.

To control for expression, we restricted the analysis to ORFs with Ribo-RET peaks (expressed genes). In this set, the small discrepancy disappears:

- Any SD motif % by start codon: ~90% for all three start codons.
- Extended SD usage % by start codon: ~30% for AUG/GUG/UUG (all within a few percent).

Importantly, discarding all annotated UUG start codons without Ribo-Ret expression peaks, still results in a substantially larger fraction of remaining UUG-starting genes in *S. aureus* compared to *E. coli* (approximately 7% compared to 2%). A note has been added in the main text to indicate that the 9% UUG start codon use from annotations alone could overestimate the actual percentage, now stating:

'Notably, this fraction may slightly overestimate the actual number of UUG start codon use, which is approximately 7% when considering only annotated ORFs with Ribo-Ret RPF expression peaks.'

We have added this newly obtained information to the main text, describing:

'Considering only ORFs expressed in our Ribo-Ret datasets, no strong difference was observed for extended or general SD motif usage between AUG, GUG and UUG start codons. In this context, general SD motif usage increased to approximately 90% while extended SD motif usage slightly increased above 30%.'

Figure R5: SD motif usage by start codon identity. Left: all annotated ORFs. Right: ORFs with Ribo-RET peaks (expressed subset). Orange, extended SD; green, any SD motif. Note the convergence across start codons in the expressed subset.

10- Figure 3F: Do genes with no or short SD sequences have distinct features? For example, could reduced secondary structure around the start codon promote efficient SD-independent initiation? A brief analysis or comment on this would be helpful.

We appreciate the suggestion and examined whether ORFs lacking a clear SD motif display distinguishing 5'-UTR features.

Sequence content. We generated a sequence logo of the 20-nt region upstream of start codons for genes without a predicted SD motif. We did not detect a single dominant motif, but we observed A-enrichment and a relative depletion of G sequence stretches, consistent with a possible lower propensity for stable structures and with potential S1-mediated affinity for single-stranded A/U-rich RNA (Fig. R6).

Figure R6: Sequence logo of *S. aureus* 5' UTRs lacking a predicted SD motif. Logo of the 20-nt region immediately upstream of annotated start codons for genes without a canonical SD sequence (n = 430). The A-enrichment and relative paucity of G runs are consistent with modestly reduced folding and potential S1 affinity.

Local secondary structure. We computed minimum free-energy (MFE) profiles with ViennaRNA (40-nt windows, 5-nt step size) from -100 to +150 nt around start codons. As a control, we performed the same analysis on *E. coli* K-12, which recapitulated the reported decrease in folding propensity near start codons (e.g., Saito et al., 2002; Del Campo et al., 2015 and Burkhardt et al., 2017). In contrast, *S. aureus* displayed overall lower global propensity for stable structures and no pronounced change in local MFE around start sites. Genes without SD motifs did not display markedly lower folding than SD-containing genes (Fig. R7). We interpret this as a consequence of the A/T-rich genome architecture: overall reduced folding likely makes strong SD-aSD pairing (including extended pairing) a primary determinant for recruitment, whereas SD-poor leaders may rely on modest A-rich single-strandedness and accessory factors (e.g., S1) rather than a universal structural signature.

References:

Saito K, Green R, Buskirk AR. Translational initiation in *E. coli* occurs at the correct sites genome-wide in the absence of mRNA-rRNA base-pairing. *eLife*. 2020;9:e55002.

Del Campo C, Bartholomäus A, Fedyunin I, Ignatova Z. Secondary Structure across the Bacterial Transcriptome Reveals Versatile Roles in mRNA Regulation and Function. *PLOS Genetics*. 2015;11(10):e1005613.

Burkhardt DH, Rouskin S, Zhang Y, Li G-W, Weissman JS, Gross CA. Operon mRNAs are organized into ORF-centric structures that predict translation efficiency. *eLife*. 2017;6:e22037.

Figure R7: Local RNA secondary structure profiles around start codons. Sliding-window MFE (ViennaRNA; 40-nt windows, 5-nt step size) from -100 to +150 nt relative to the start codon. Top: Genome-wide comparison of *S. aureus* HG001 and *E. coli* K-12 confirms a start-site MFE change in *E. coli* but not in *S. aureus*. Middle: *S. aureus* genes with vs. without predicted SD motifs show similar MFE profiles. (Bottom) *S. aureus* genes grouped by start codon (AUG/GUG/UUG) show comparable MFE behavior.

11- Figure 5: The authors report 17 uORFs, of which 8 were tested by toeprinting and 5 by luciferase assay. Could the authors clarify the criteria used to select uORFs for each experiment? This would help readers understand the rationale behind the experimental choices.

We appreciate the request to clarify selection criteria. From the 17 uORF candidates, we prioritized uORFs based on a combination of features:

1. Regulatory plausibility at the TIS: close proximity/overlap with the downstream RBS/start codon (e.g., *rbfL*, *gehA*), which increases the likelihood of RBS occlusion–based control.
2. Physiological/virulence relevance: downstream genes with known roles in *S. aureus* biology (e.g., *rbf* for biofilm formation; prophage lysis modules for sORF15/16).
3. Signal strength and definition in Ribo-RET: sharp initiation peaks at the uORF start (high S/N), suggesting active initiation suitable for toeprinting.
4. Sequence features suggestive of regulation: rare codons or features that could couple translation to nutrient status (e.g., Arg codons in *rbfL* or the multiple Asn codons in sORF7), or intrinsic terminator motifs following the uORF (sORF15/16).

Using these criteria, we tested 8 uORFs by toeprinting (prioritizing clear initiation signatures and proximity to the downstream TIS) and advanced five of them to *in vivo* luciferase assays (prioritizing physiological relevance and reporter tractability). The *rbfL* uORF was pursued in depth because it was highly expressed *in vivo* and controls Rbf, a transcription factor promoting biofilm formation.

We have now indicated our selection criteria for experimental validation in the Results section as follows:

‘These uORFs were selected for experimental validation based on four criteria: (i) proximity or overlap with the downstream TIS, (ii) virulence or physiological relevance of the downstream gene, (iii) strength/sharpness of Ribo-RET initiation peaks, and (iv) the presence of sequence features indicative of conditional control, such as rare codons, or intrinsic terminators.’

12- Figure 6C–E: Cross-comparison between panels is difficult due to the different y-axis scales.

We agree that direct cross-panel comparison is hindered by differing y-axis scales. Because panels C–F report distinct media conditions (BHI vs RPMI) with different absolute reporter outputs, we keep condition-specific axes to avoid misleading normalization. To facilitate comparison, we now include a summary panel (new Fig. 6G) showing fold-change with and without the *rbfL* start codon within each condition, enabling direct cross-condition interpretation.

Despite that, could the authors clarify why the *rbf*-uORF mutant behaves differently in RPMI with or without 10× L-Arg?

The different behavior of the *rbf* uORF mutant in RPMI ± 10× L-Arg follows from our model that Arg availability controls charging of the low-abundance tRNA^{Arg(UCU)}, which decodes both AGA/AGG codons in *rbfL*. In poor RPMI, reduced tRNA^{Arg(UCU)} charging slows decoding at these sites, causing co-translational pausing on *rbfL* that occludes the *rbf* RBS and represses *rbf* initiation. Supplementing 10× L-Arg replenishes the amino-acid pool, restores tRNA charging, and alleviates pausing, thereby derepressing *rbf*. Consistently, overexpressing tRNA^{Arg(UCU)} yields a partial rescue, as it improves competition for aminoacylation but cannot fully restore charging under limitation.

We now clarify this mechanism in the text and support it with newly added co-translational toeprinting experiments: Pauses occur at position +13 in respect to the AGA/AGG codons *in vitro* and disappear upon Ala-scan substitutions at these positions (new Supplementary Fig. S6A-B). These toeprint positions are consistent with slow decoding (pausing) of the respective codons in the ribosomal A-site.

To further clarify, we have now revised the Discussion to explicitly highlight the role of tRNA charging levels:

“We show that a novel leader peptide, rbfL, senses arginine limitation through the presence of rare arginine codons, which are decoded by the low-abundance tRNA^{Arg(UCU)}. When arginine is scarce and tRNA charging levels decrease, ribosomes pause on rbfL and block the ribosome binding site of the downstream gene rbf, preventing its efficient translation initiation.”

This is supported by our reporter assays supplementing L-arginine or over-expressing the tRNA^{Arg(UCU)} in the poor medium, relieving repression of *rbf*. Our rationale for this interpretation is that tRNA charging levels, including tRNA^{Arg(UCU)}, decrease under nutrient limitation. Subsequently, the two Arginine codons within *rbfL* are slowly decoded and pause ribosomes co-translationally. Our own nanopore-based tRNA quantification of *S. aureus* (Pre-print: Jaramillo-Ponce et al. 2025) revealed tRNA^{Arg(UCU)} as a low abundant tRNA isoacceptor species. Under nutrient limiting conditions, it competes with the other tRNA isoacceptors for charging. On one hand, replenishing the pool of available L-

Arginine is expected to fully restore charging levels. On the other hand, increasing the levels of the tRNA^{Arg(UCU)}, is expected to improve its ability to compete with the more abundant tRNA isoacceptors and to increase the amount of charged tRNA^{Arg(UCU)} in the cell. Indeed, such effects are well documented, including for rare arginine codons, where it could be shown that *argU* (encoding tRNA^{Arg(UCU)}) co-expression can improve recombinant protein expression in *E. coli*, as for example shown by Garcia et al. (1996).

References:

Garcia OL, González B, Menéndez A, Sosa AE, Fernández JR, Santana H, et al. The *argU* gene product enhances expression of the recombinant human alpha 2-interferon in *Escherichia coli*. *Ann N Y Acad Sci.* 1996;782:79-86.

Jaramillo-Ponce JR, Wolff P, Marchand V, Motorin Y, Kohl M, Kanazawa H, et al. Complete post-transcriptional modification profiles in individual *Staphylococcus aureus* tRNA species. *bioRxiv.* 2025:2025.10.30.685614.

Also, wouldn't we expect *rbfL* expression to be similar in RPMI + 10× L-Arg and RPMI + tRNA^{Arg}?

As mentioned above, we do not expect identical *rbfL* expression in RPMI + 10× L-Arg and RPMI + tRNA^{Arg(UCU)}, because these manipulations act at different points:

- 10× L-Arg replenishes the amino-acid pool and fully restores charging of tRNA^{Arg(UCU)}, thereby alleviating *rbfL* pausing and derepressing *rbf*.
- tRNA^{Arg(UCU)} overexpression improves competition for aminoacylation but does not increase Arg availability; therefore, charging recovery is partial, and so is the relief of pausing/derepression.

A second, practical point is that our bicistronic normalization (Firefly/Renilla) is optimal for within-condition comparisons (e.g., \pm *rbfL* start codon), but cross-condition comparisons (\pm tRNA overexpression) are confounded because both reporters contain Arg codons; the normalization itself can shift with Arg limitation or tRNA^{Arg(UCU)} levels. For that reason, we present data condition-wise and emphasize fold-changes (new Fig. 6G).

13- Figure 6G: what is the effect of the amino acid substitution on the expression of *rbfL*?

We generated an *rbfL* reporter with the R→G substitution (AGG→GGG) and measured its expression in RPMI. The mutant *rbfL*-fluc showed ~40% of the translation efficiency observed for the WT *rbfL* in the same condition. As detailed in our response to Reviewer 1 (point 6), changes in this region are highly constrained because both Arg codons contribute to the SD motif for the downstream *rbf* TIS. Consistent with this, the

AGG→GGG change uncouples *rbf* expression from *rbfL* and Arg availability, *rbf* translation increases irrespective of *rbfL*, indicating that the nucleotide change improves *rbf* initiation rather than specifically testing the role of the rare codon. We therefore complemented reporter data with co-translational toeprinting (new Fig. S6A–B), which directly shows pausing over the AGA/AGG positions and its loss upon Ala substitutions. For completeness, we also tested AGG→GAG (Glu), which similarly uncouples *rbf* from *rbfL* (Fig. S6D). Jointly, these results support the model that *rbfL* regulates *rbf* via Arg-sensitive pausing and RBS occlusion, while also illustrating the fine-tuning of the shared *rbfL/rbf* initiation architecture.

14- Figure 6H: The authors propose that the 30S cannot initiate due to a stalled 70S on *rbfL*. However, previous studies have demonstrated “70S scanning initiation,” where post-termination 70S ribosomes can scan downstream for an SD sequence. Could the authors comment on this alternative mechanism and whether it may apply in this context?

We appreciate the suggestion to consider 70S scanning/termination–re-initiation (TeRe). The *rbfL*→*rbf* architecture (4-nt spacing; **UAAAGUUAUG**) could, in principle, permit TeRe, and prior work by Huber et al. (2019), shows that a downstream SD can support re-initiation while also recruiting 30S for canonical *de novo* initiation. In our case, multiple lines of evidence indicate that *de novo* 30S initiation predominates:

- Toeprinting (Fig. 6B) shows robust 30S initiation complexes at both *rbfL* and *rbf* start sites, demonstrating that *rbf* is not strictly translationally coupled to *rbfL* through TeRe.
- In rich medium, converting the *rbfL* start codon (GUG→UAG) has minimal impact on *rbf* reporter output, arguing against a dominant TeRe contribution.
- Under arginine limitation, *rbfL* pausing occludes the *rbf* RBS, which would inhibit both *de novo* 30S initiation and any potential post-termination 70S scanning access, consistent with the observed repression.

While we cannot exclude a minor TeRe component, our data support a model in which Arg-sensitive pausing on *rbfL* reduces access of 30S to the *rbf* RBS, and *de novo* initiation is the principal route.

Reference:

Huber M, Faure G, Laass S, Kolbe E, Seitz K, Wehrheim C, et al. Translational coupling via termination-reinitiation in archaea and bacteria. *Nature Communications*. 2019;10(1):4006.

15- Figure 6: The authors propose that **ribosome stalling** on a small uORF reduces translation of the downstream gene *rbf*, which is involved in biofilm formation. Has this hypothesis been tested by **assessing biofilm formation in nutrient-poor media**?

We agree this is an important test. Because absolute biofilm output is intrinsically lower in poor media, we specifically assayed RPMI ± L-Arg to probe the model's prediction that arginine availability relieves *rbfL*-dependent repression of *rbf*. Overnight cultures were diluted 1:100 into RPMI (± 10× L-Arg), grown 48 h static at 37 °C in 96-well plates (n = 24 technical replicates/condition, two independent experiments), planktonic growth was measured from the supernatant, and biofilm was fixed (MeOH) and stained (crystal violet). L-Arg supplementation significantly increased biofilm formation while reducing planktonic OD₆₀₀ measurements, indicating a specific promotion of biofilm, and not a general growth improvement (new Supplementary Fig. S6C).

These data are consistent with our model: Arg limitation lowers charging of tRNA^{Arg(UCU)}, enhancing *rbfL* pausing and RBS occlusion at *rbf*. L-Arg addition restores charging, derepresses *rbf* translation, and increases biofilm. They also align with prior links between Arg metabolism and *S. aureus* biofilm, for example the conserved sRNA RsaE was found to repress Arg catabolism by Rochat et al. (2018), while promoting biofilm formation as shown by Schoenfelder et al. (2019). Of particular note, Arg restriction was recently demonstrated to limit growth in *S. aureus* biofilms, resulting in antibiotic tolerance of the bacteria (Freiberg et al., 2024). These points have been mentioned in the discussion.

References:

Rochat T, Bohn C, Morvan C, Le Lam TN, Razvi F, Pain A, et al. The conserved regulatory RNA RsaE down-regulates the arginine degradation pathway in *Staphylococcus aureus*. *Nucleic Acids Res.* 2018;46(17):8803-16.

Schoenfelder SMK, Lange C, Prakash SA, Marincola G, Lerch MF, Wencker FDR, et al. The small non-coding RNA RsaE influences extracellular matrix composition in *Staphylococcus epidermidis* biofilm communities. *PLOS Pathogens.* 2019;15(3):e1007618.

Freiberg JA, Reyes Ruiz VM, Gimza BD, Murdoch CC, Green ER, Curry JM, et al. Restriction of arginine induces antibiotic tolerance in *Staphylococcus aureus*. *Nature Communications.* 2024;15(1).

16- The proposed ribosome stalling at the uORF is intriguing. Have the authors considered Ribo-seq in poor media to explore this? Not essential, but it could offer useful insights.

We agree that performing Ribo-seq in nutrient-poor conditions would be highly informative. Such datasets could (i) reveal initiation events and *loci* that are weak or silent in rich media, and (ii) uncover charging-sensitive translational pauses that emerge under global nutrient stress, potentially extending beyond *rbfL* to other short uORFs (e.g., the Asn-rich sORF7). More broadly, poor-media Ribo-seq would help delineate translational adaptations to nutrient limitation, including possible shifts in initiation control, elongation dynamics, and ORF usage.

That said, nutrient-poor growth substantially alters ribosome occupancy, footprint yield, and nuclease sensitivity, necessitating dedicated protocol re-optimization. Given these practical and scope constraints, we view poor-media Ribo-seq as a promising direction for future work. In the present study, we instead consolidate the mechanism through co-translational toeprinting that shows pauses within *rbfL*, and *in vivo* reporter assays showing nutrient/condition-dependent regulation.

17- Figure S3: The start codon from the old annotation is not marked. Could the authors zoom out or adjust the figure to indicated the position of the old start codon for comparison?

We are grateful for the suggestions to improve figure clarity. We have now updated Supplementary Figure S3 to explicitly mark the previously annotated start codon alongside the reannotated start codon, displaying the full sequence context of the reannotations.

18- Figure S5: In *E. coli*, *infC* initiates from a non-canonical start codon (AUU). Does it also have an extended SD sequence, or is this feature unique to *Staphylococcus*? Similarly, for the other conserved genes with non-canonical start codons, does the link to extended SD motifs occur only in *Staphylococcus*, or is it more broadly conserved?

For *E. coli infC* (AUU start), there is no evidence of an extended SD-aSD helix of the type we describe in *S. aureus*. Our biochemical comparisons and the new cryo-EM data indicate that *E. coli* 70S does not accommodate upstream, start codon-proximal extensions of the SD-aSD helix, consistent with the position/density of bS21 on the 30S platform and the lack of structural reports of extended pairing in *E. coli*. Accordingly, non-canonical initiation in *E. coli* (e.g., *infC* AUU) proceeds without such extended SD pairing.

By contrast, in *S. aureus* we find that novel non-canonical starts (e.g., AUU, AUA) are linked to extended SD-aSD interactions (as many canonical AUG and GUG starts), which support P-site placement and decoding. We expect this architecture to be more broadly present in AT-rich Firmicutes (and potentially other clades showing longer average SD-

start spacing, e.g., the Cyanobacterium *Synechocystis*), as suggested by cross-species spacer analyses reported by Ma et al. (2002).

Reference:

Ma J, Campbell A, Karlin S. Correlations between Shine-Dalgarno Sequences and Gene Features Such as Predicted Expression Levels and Operon Structures. *Journal of Bacteriology*. 2002;184(20):5733-45.

Minor comments:

1- Fig. 1G: this representation suggests that STAR sORFs may have a “negative length”.

We thank the reviewer for noting this. We have revised Fig. 1G to avoid any visual implication of “negative length” by plotting individual data points within violin plots anchored at zero, with explicit zero baselines and axis labels. The figure legend was updated accordingly to clarify the display.

2- Line 33: Consider introducing the abbreviation "S. aureus" when first mentioning *Staphylococcus aureus*.

3- The usage of "*Staphylococcus aureus*" and "S. aureus" is inconsistent throughout the text. Please standardize the abbreviation after its first introduction

We have updated the manuscript to introduce “*S. aureus*” at its first mention of *Staphylococcus aureus* and standardized the abbreviation throughout for consistency.

4- Line 230: reference to Fig. 3 A, D should be Fig. 3 C, D

We thank the reviewer for catching this. The text now correctly cites the original Fig. 3C, D (new Fig. 3D, E in the revised manuscript), and all figure panel references were double-checked for consistency throughout the manuscript.

Reviewer #3 (Remarks to the Author):

Reviewer #4 (Remarks to the Author):

The paper of Kohl et al. explores the principles of translation initiation and start codon selection by the ribosome in *Staphylococcus aureus*. To this end, the authors utilized Ribo-seq analysis of the cells treated with retapamulin, a drug that selectively stalls translating ribosomes at start codons (Ribo-Ret). The Ribo-Ret was previously used to map start sites in *E. coli*. However, in this case, the digestion of the polysome fraction with MNase, a nuclease with a strong sequence bias, limits precise mapping of ribosome-protected fragments and therefore the identification of start sites (especially in cases where several potential start sites are close to each other). In this paper, the authors used RNase I instead of MNase, which allowed them to significantly improve the quality of mapping of initiation sites in *S. aureus*. Further analysis of Ribo-Ret data, combined with structural studies, showed that this bacterium relies on a specific SD-aSD interaction to select the start codon, which is longer than the corresponding interaction in *E. coli*. This conclusion is supported by an elegant in vitro toe-print analysis. The authors further demonstrate that such extended SD-aSD interaction can be utilized by *S. aureus* (and likely other Firmicutes) for uORF-mediated translational control of downstream gene expression, including those that regulate biofilm formation in this human pathogen.

This is an exciting and illuminating study that significantly extends our understanding of translation initiation in bacteria. The experiments are very 'clean', convincing, and properly controlled. The paper is well-written and an easy read. The study represents novel findings and is appropriate for publication.

We sincerely thank the reviewer for the thoughtful, positive assessment. We are glad that the advance, using RNase 1-based Ribo-RET for precise initiation mapping, together with our cryo-EM and toeprinting data, came across clearly, and that the extended SD-aSD mechanism and uORF-mediated control (*rbfL*→*rbf*) were compelling. Our study indeed provides novel insights into the specific nature of *S. aureus* translation initiation.